# Dynamic MAIT cell response with progressively enhanced innateness during acute HIV-1 infection

Kerri G. Lal[1,2,3], Dohoon Kim [1,2], Margaret C. Costanzo[1,2], Matthew Creegan[1,2], Edwin Leeansyah[3,4], Joana Dias[3], Dominic Paquin-Proulx [1,2], Leigh Anne Eller[1,2], Alexandra Schuetz[1,2,5], Yuwadee Phuang-ngern[5], Shelly J. Krebs [1,2], Bonnie M. Slike[1,2], Hannah Kibuuka[6], Lucas Maganga[7], Sorachai Nitayaphan[8], Josphat Kosgei[9], Carlo Sacdalan [10], Jintanat Ananworanich [1,2,10], Diane L. Bolton[1,2], Nelson L. Michael[1], Barbara L. Shacklett [11], Merlin L. Robb[1,2], Michael A. Eller [1,2] & Johan K. Sandberg [3]*

Mucosa-associated invariant T (MAIT) cell loss in chronic HIV-1 infection is a significant insult to antimicrobial immune defenses. Here we investigate the response of MAIT cells during acute HIV-1 infection utilizing the RV217 cohort with paired longitudinal pre- and post-infection samples. MAIT cells are activated and expand in blood and mucosa coincident with peak HIV-1 viremia, in a manner associated with emerging microbial translocation. This is followed by a phase with elevated function as viral replication is controlled to a set-point level, and later by their functional decline at the onset of chronic infection. Interestingly, enhanced innate-like pathways and characteristics develop progressively in MAIT cells during infection, in parallel with TCR repertoire alterations. These findings delineate the dynamic MAIT cell response to acute HIV-1 infection, and show how the MAIT compartment initially responds and expands with enhanced function, followed by progressive reprogramming away from TCR-dependent antibacterial responses towards innate-like functionality.

[1] U.S. Military HIV Research Program, Walter Reed Army Institute of Research, Silver Spring, MD, USA. [2] Henry M. Jackson Foundation for the Advancement of Military Medicine, Bethesda, MD, USA. [3] Center for Infectious Medicine, Department of Medicine, Karolinska Institutet, Stockholm, Sweden. [4] Program in Emerging Infectious Diseases, Duke-National University of Singapore Medical School, Singapore, Singapore. [5] Department of Retrovirology, Armed Forces Research Institute of Medical Sciences, Bangkok, Thailand. [6] Makerere University Walter Reed Project, Kampala, Uganda. [7] National Institute for Medical Research-Mbeya Medical Research Center, Mbeya, Tanzania. [8] Royal Thai Army Component, Armed Forces Research Institute of Medical Sciences, Bangkok, Thailand. [9] Kenya Medical Research Institute/U.S. Army Medical Research Directorate-Africa/Kenya, Kericho, Kenya. [10] SEARCH, The Thai Red Cross AIDS Research Centre, Bangkok, Thailand. [11] Department of Medical Microbiology and Immunology, School of Medicine, University of California Davis, Davis, CA, USA. *email: johan.sandberg@ki.se

Mucosa-associated invariant T (MAIT) cells are an evolutionarily conserved subset of unconventional T cells, highly abundant in mucosal tissues, peripheral blood, and the liver of humans[1–3]. MAIT cells express a semi-invariant αβ T cell receptor (TCR)[4–6], and recognize microbial vitamin B2 metabolite antigens from a wide range of bacteria and fungi presented by the major histocompatibility complex (MHC) class I-related (MR) 1 molecules[7,8]. MAIT cells activated by MR1-presented antigens respond rapidly with release of cytokines including IFNγ, TNF, and IL-17[1,9], and mediate cytolytic function against bacterially infected cells[10–12]. Their response pattern is dependent on a transcriptional profile characterized by the co-expression of promyelocytic leukemia zinc finger (PLZF), and retinoid-related orphan receptor (ROR) γt[1,2,13].

The unique ability of MAIT cells to respond to conserved bacterial-derived and fungal-derived metabolites is important for protection against microbial infections, in particular mycobacterial and other infections of the lung[9,14–18]. In addition, high expression of the receptors for IL-18 and IL-12 provides MAIT cells with the capacity to respond to antigen-presenting cell (APC)-derived cytokines[9], recently shown to be important for enhancement of TCR-mediated MAIT cell activation[19,20], and for triggering of MR1-independent MAIT cell responses[21–23]. Such MR1-independent responses, including production of IFNγ, may be important for the involvement of MAIT cells in viral diseases[24–28]. Interestingly, recent findings in murine models suggest that MAIT cells may play a role in limiting viral replication and immunopathogenesis of influenza virus infection[29]. Thus, MAIT cells are poised to respond to infection from a variety of pathogens and can possibly influence disease outcome.

The impact of chronic HIV-1 infection on MAIT cells has been investigated (reviewed in ref. [30]), with declining MAIT cell frequency and function in response to in vitro antigen exposure, in cross-sectional studies of untreated infection[24,25,27]. Combination anti-retroviral treatment (cART) partly restores MAIT cell function, but their numerical decline appears irreversible in the blood[24,25]. The basis for MAIT cell loss is unclear, but may involve recruitment to inflamed mucosa[31]. The gut mucosa is a central site in HIV immunopathogenesis where macrophages and T cells are recruited and mediate an inflammatory cytokine storm in the earliest days after infection[32,33]. The peak of viral replication occurs around two weeks after infection, followed by recession to a set-point level approximately one month post-infection[34].

Mucosal sites are of critical importance throughout the natural course of HIV-1 infection. Impaired integrity of the gut mucosal barrier with translocation of microbes and microbial products into the underlying tissues and circulation is believed to contribute strongly to immune activation, inflammation, and accelerated disease[35]. Mucosal immunity is impaired in HIV-infected subjects, with severe consequences for control of important pathogens such as *M. tuberculosis*, as well as other microbes that encode the riboflavin biosynthesis pathway. Interestingly, mice deficient in MR1, thus lacking MAIT cells, display signs of impaired gut integrity and increased microbial translocation[36]. Mucosal MAIT cells express the tissue-protective cytokine IL-22 suggesting a broader role in protection of the mucosa[37,38]. While it is well established that MAIT cells decline in chronic stages of HIV-1 infection, their dynamics and response during acute HIV-1 infection have yet to be elucidated. Such studies are critical to understand the role of MAIT cells in HIV-1 immunopathogenesis and impaired antimicrobial immunity in the infected human host. In this study, we investigate the response of MAIT cells during the first critical days and weeks of acute HIV-1 infection utilizing longitudinal pre-infection and post-infection samples from the RV217 Early Capture HIV Cohort Study (ECHO)[34].

Already at the time of peak HIV-1 viremia, MAIT cells become activated and expand in both blood and mucosa in a manner associated with markers of microbial translocation. This is followed by enhanced MAIT cell function around the time of set-point viral load establishment, and later by their functional impairment in chronic stages of infection. Notably, MAIT cells develop enhanced innate-like transcriptional and phenotypic characteristics progressively over time during infection. Thus, the MAIT cell compartment responds in a dynamic fashion to acute HIV-1 infection with initial expansion and enhanced function, followed by a progressive shift away from TCR-dependent antimicrobial responses towards innate-like functional characteristics.

## Results

**Early MAIT cell expansion during acute HIV-1 infection.** MAIT cell dynamics were examined during the earliest stages of acute HIV-1 infection in 29 individuals from the RV217 ECHO study[34], for which cryopreserved autologous longitudinal samples were available from pre-infection time points, followed by samples taken within days from the first detection of HIV-1 RNA and further out into chronic infection (Supplementary Table 1). MAIT cells, identified by surface expression of TCR Vα7.2 and CD161 (Fig. 1a and Supplementary Fig. 1), were measured in samples available from up to ten time points per donor spanning nearly three years after infection (Fig. 1b). The relative frequency of MAIT cells among CD3+ T cells did not change significantly over the course of acute HIV-1 infection. However, given the broad T cell subset redistribution occurring in the blood during acute HIV-1 infection[34], we next analyzed the absolute counts of MAIT cells and observed a pattern of increasing MAIT cells counts during acute infection (Fig. 1c). This trend reached significance for donors where paired time points were available from a median (range) 23 (8–67) MAIT cells/μl in early infection (median 1 day since first positive test for HIV-1 RNA), to 27 (14–97) MAIT cells/μl by day 43 post-infection ($p = 0.016$; Wilcoxon Signed Rank test) (Fig. 1d). This increase coincided with the bulk CD8 T cell expansion two weeks after peak viremia, and before viral load set-point was established (Fig. 1c). MAIT cell counts later returned to baseline levels by early chronic infection. However, a trend was observed 600 days after the first detectable HIV-1 nucleic acid test where MAIT cell absolute counts declined to a median (range) of 18 (8–38) cells/μl, below levels observed at baseline. The identification of MAIT cells using the combination of Vα7.2 and CD161 expression among CD3+ cells was confirmed using the MR1–5-OP-RU tetramer (Fig. 1a, Supplementary Fig. 1 and Supplementary Fig. 2).

To investigate changes in MAIT cells in tissues during acute HIV-1 infection, we examined cells isolated from rectal mucosal biopsies from 7 acutely HIV-1 infected individuals sampled before initiation of cART and 17 uninfected matched controls from the RV254 and RV304 cohorts using flow cytometry[39] (Supplementary Table 2). In this cross-sectional data set, relative rectal mucosal MAIT cell levels were significantly higher in the acutely infected subjects as compared to the uninfected controls (Fig. 1e). This pattern reflected an absolute increase in MAIT cells per gram of rectal tissue (Fig. 1f), supporting the notion that the temporary expansion of MAIT cells observed in peripheral blood also occurs in gut mucosa.

The majority of human MAIT cells express CD8, with a smaller subset being CD8 and CD4 double-negative (DN) (Fig. 1a). Through acute and early HIV-1 infection there was a tendency towards subset redistribution with a slight decline in frequency of CD8+ MAIT cells, and a corresponding gain in frequency of DN MAIT cells (Fig. 1g). CD4+ MAIT cells are a minor subset of the total MAIT cell pool (Fig. 1a). Here, the MR1

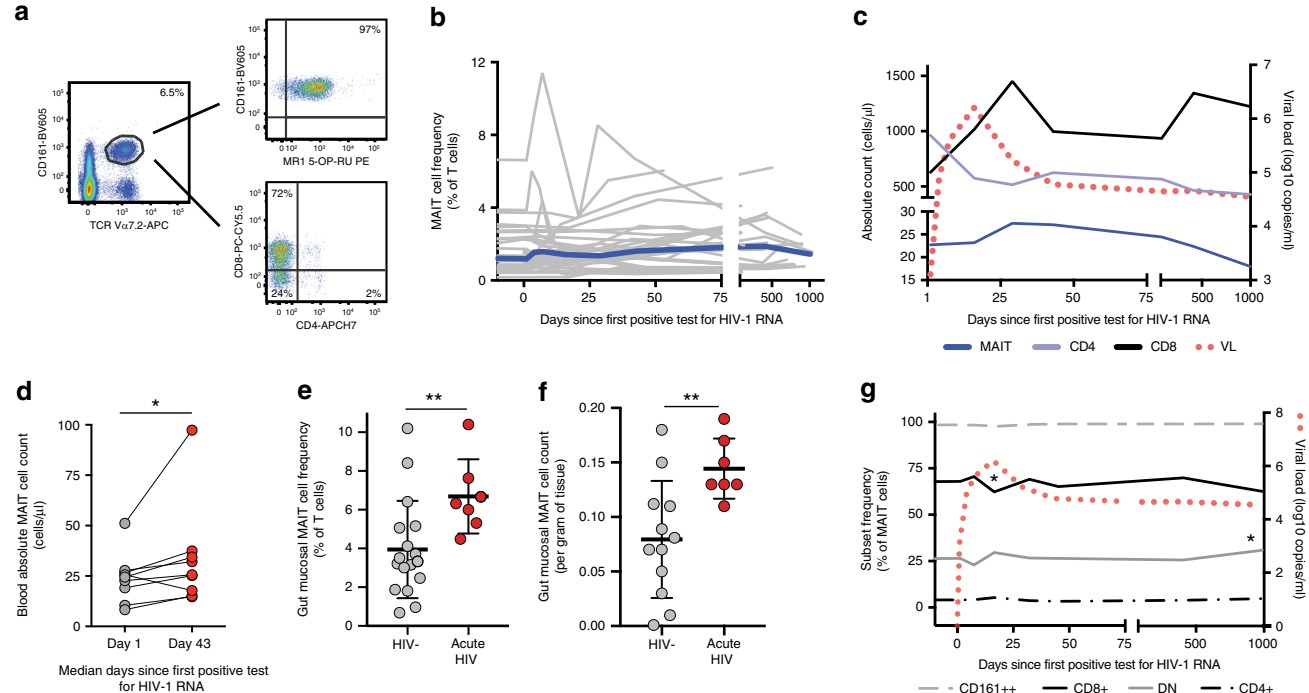

**Fig. 1 MAIT cell activation and dynamics in the RV217 acute HIV-1 infection cohort. a** Representative flow cytometry plots of a pre-infection time point from one individual enrolled in the RV217 acute capture cohort showing identification of MAIT cells as CD161++Vα7.2+ of CD3+CD14-CD19- live lymphocytes in PBMC. Confirmation of MAIT cell identification using the 5-OP-RU loaded MR1 tetramer, and the relative distribution of CD8 and CD4 positivity within the MAIT cell gate is displayed. **b** MAIT cell frequency as a percentage of CD3+ T cells is shown longitudinally in individuals with acute HIV-1 infection (gray), with median frequency (blue) (n = 29). **c** Median absolute counts (cells/μl of blood) of MAIT cells relative to conventional CD4 and CD8 T cells and HIV-1 viral load displayed over time in acute HIV-1 infection (n = 29). **d** MAIT cell absolute counts in 9 donors where matching data points were available from days 1 and 43 after HIV-1 infection. **e** MAIT cell percentages out of total CD3+ cells isolated from rectal biopsies from individuals with acute HIV infection (n = 7), and matched uninfected controls (n = 17). **f** MAIT cell count per gram of rectal biopsy tissue from individuals with acute HIV infection (n = 7), and matched uninfected controls (n = 17). **g** Median MAIT cell subset distribution displayed over time in acute HIV-1 infection. CD161, CD8, CD4, and double negative expression in MR1 tetramer-defined MAIT cells displayed (n = 19). *p ≤ 0.05, **p ≤ 0.01, ***p ≤ 0.001. In **d**, statistical analysis was performed using Wilcoxon Signed Rank test; in **e** and **f** using the Mann-Whitney test; in **g** the nonparametric Friedman test with the Dunn's multiple comparison test. PBMC, Peripheral blood mononuclear cells. MAIT cells are identified as CD161++Vα7.2+ within CD3+CD14-CD19- live lymphocytes, except in **e**, where MAIT cells are identified using the 5-OP-RU loaded MR1 tetramer of CD3+CD14-CD19- live lymphocytes. 5-OP-RU 5-(2-oxopropylideneamino)−6-d-ribitylaminouracil. VL viral load. The source data underlying **b**, **c**, and **g** are provided as a Source Data file.

tetramer-defined MAIT cell population showed a significant decline in the CD4+ subset among total MAIT cells from pre-infection to the early chronic time point (p = 0.003; Friedman test with the Dunn's multiple comparison test) (Supplementary Table 3). In contrast, expression of CD161 in the MR1 tetramer-defined MAIT cell population was unchanged and consistently high throughout the period following HIV-1 infection (Fig. 1g). Taken together, these findings reveal that in acute HIV-1 infection there is a brief period of MAIT cell expansion and maintenance, which includes significant changes in subset representation, before loss of this population commences in chronic infection.

**IRF4 expression predicts MAIT cell levels at viral set-point.** Acute HIV-1 infection is associated with strong activation of conventional T cells, and in particular CD8 T cells[40,41]. To ascertain the temporal dynamics of MAIT cell activation in acute HIV infection, we examined phenotypic markers of activation and also sorted MAIT cells for targeted transcriptomic analysis from pre-infection and three post-infection samples by flow cytometry. At peak viremia the frequencies of MAIT cells expressing HLA-DR, CD38, Programmed Death 1 (PD-1), T cell immunoreceptor with Ig and ITIM domains (TIGIT) and granzyme B (GrzB) were elevated above pre-infection frequencies, and transcripts for these proteins remained elevated above pre-

infection expression throughout acute HIV-1 infection (Fig. 2a and Fig. 2b). Similarly, expression of CCR5, already high at the resting state, increased significantly in MAIT cells during acute infection (Supplementary Table 3 and Supplementary Fig. 2). Transcriptional analysis further revealed that transcripts encoding the proliferation-specific protein Ki67 (*MKI67*), and the transcription factor interferon regulatory factor 4 (*IRF4*) were elevated at peak viremia compared to baseline (p = 0.05 and p = 0.03, respectively; Friedman test with the Dunn's multiple comparison test) (Fig. 2b). Interestingly, in parallel to the magnitude of HIV-1 viremia, MAIT cell expression of activation markers, including the immune checkpoint receptor PD-1 (Fig. 2a), correlated positively with both *IRF4* and *MKI67* gene expression (Fig. 2c and Fig. 2d). By day 85 the *MKI67* expression had returned to levels observed at baseline, whereas the *IRF4* transcript continued to be significantly elevated (p = 0.04; Friedman test with the Dunn's multiple comparison test) (Fig. 2b). This was in contrast to KI67 protein expression measured at the same time point, which remained elevated compared to pre-infection (Supplementary Table 3). Early into chronic infection, the frequency of MAIT cells expressing PD-1 correlated inversely with MAIT cell frequency (Supplementary Fig. 3). Furthermore, levels of *IRF4* mRNA expression at peak viremia correlated inversely with MAIT cell counts (Fig. 2e), and frequency (Fig. 2f), at the time of viral load set-point and into early chronic infection. Thus,

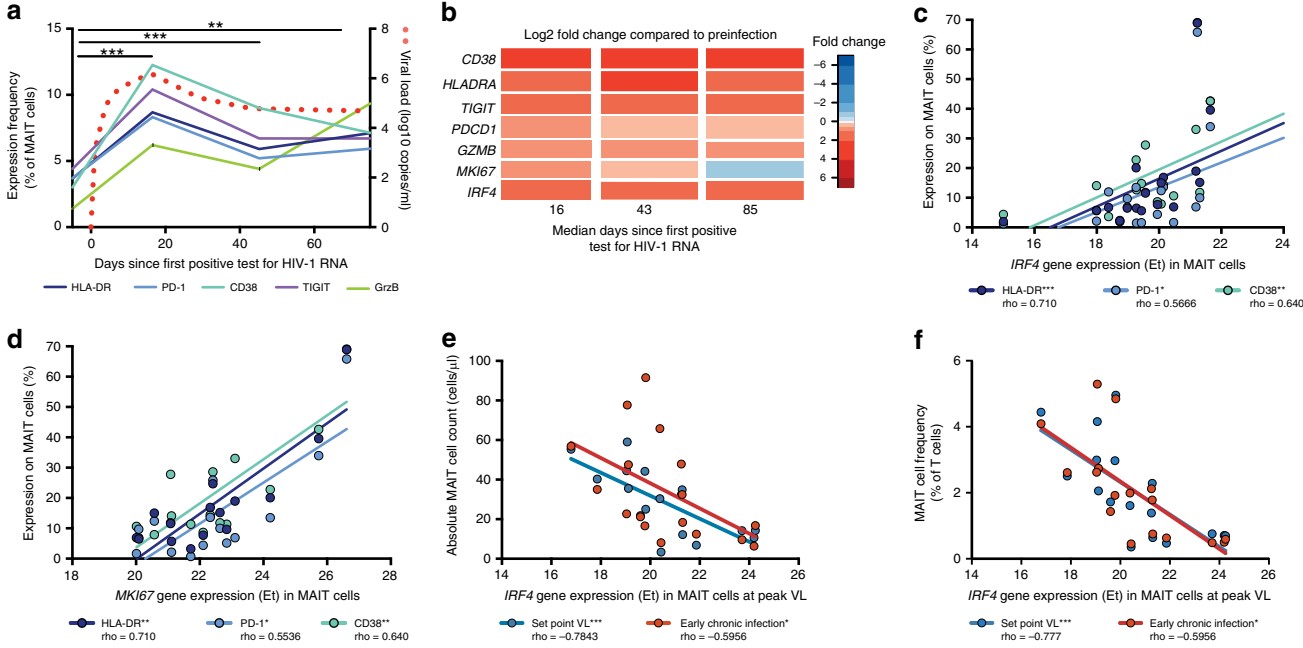

**Fig. 2 MAIT cell activation in acute HIV-1 infection. a** Median expression of markers of activation and exhaustion (HLA-DR, PD-1, CD38, TIGIT, and GrzB) in MAIT cells in PBMC as assessed by flow cytometry displayed over time in acute HIV-1 infection ($n = 19$). **b** The fold change of gene expression compared to pre-infection of individual genes (*CD38, HLADRA, TIGIT, PDCD1, GZMB, MKI67,* and *IRF4*) in three post-infection time points in acute HIV-1 infection from bulk sorted MAIT cells ($n = 20$). **c** Correlation of *IRF4* and **d**, *MKI67* gene expression in bulk sorted MAIT cells with the protein expression of markers activation (HLA-DR, PD-1, and CD38) at the post-infection time point corresponding with peak VL (median 16 days since first positive test for HIV-1 RNA) ($n = 20$). **e** Correlation of gene expression of *IRF4* in sorted MAIT cells with MAIT cell absolute counts, or **f**, MAIT cell frequency at two post-infection time points corresponding with set point VL (median 43 days since first positive test for HIV) or early chronic infection ($n = 20$). *$p \le 0.05$, **$p \le 0.01$, ***$p \le 0.001$. In **a**, statistical analysis was performed using the nonparametric Friedman test with the Dunn's multiple comparison test; in **c, d, e,** and **f** correlative analyses were performed using Spearman Rank correlation test. In **a**, significance indicated is valid for all markers displayed. PBMC, Peripheral blood mononuclear cells. MAIT cells are identified as CD161 + + Vα7.2 + within CD3 + CD14-CD19- live lymphocytes. VL viral load. The source data underlying **a** and **b** are provided as a Source Data file.

the initial upregulation of *MKI67* transcription is consistent with a period of activation-induced proliferation, whereas the induction and maintenance of *IRF4* is associated with the subsequent reduced frequency of MAIT cells.

**MAIT cell transcriptional dynamics in acute HIV-1 infection.** To gain a better understanding of the MAIT cell response to HIV-1 infection, MAIT cells were sorted from PBMCs of nine donors at pre-infection, peak viremia, viral load set-point, and early into chronic infection. Isolated MAIT cells were subjected to whole-genome transcriptional analysis using RNA-sequencing (RNA-Seq). The RNA-Seq data revealed dynamic transcriptional changes in MAIT cells compared to donor-matched data from the pre-infection time point (Fig. 3a). At peak HIV-1 viremia, expression of 61 genes was significantly upregulated and expression of 72 genes was downregulated (Fig. 3b and Supplementary Table 4). As the viral load receded to a set-point level, approximately 43 days after detectable HIV-1, the transcriptional changes were dominated by reduced expression of 123 gene transcripts, and increased expression of 24 genes relative to pre-infection. Finally, at the early chronic stage, transcriptional activation was further reduced with fewer changes observed compared to the pre-infection time point. Analysis of these data revealed a highly dynamic pattern of gene expression in MAIT cells during acute HIV-1 infection with little overlap between time points (Fig. 3c and Supplementary Table 4).

Of the top 10 transcripts upregulated at peak HIV-1 viral load, the majority (6 out of 10) were related to cell cycle and cell division including *RRM2, MYBL2, CDK1, UBE2C, CDC45,* and

*TK1* with 8 to 15-fold increased expression compared to the pre-infection samples (Supplementary Table 4). Similarly, at this time point the transcript for an inhibitor of apoptosis, *BIRC5* was increased 8-fold compared to the pre-infection expression level. The majority of cell cycle gene transcripts, including *RRM2, MYBL2, CDK1,* and *TK1,* remained significantly elevated at early set-point viral load time points, as did *BIRC5*. At the chronic stage of infection the majority of cell cycle gene transcripts returned to pre-infection levels, except for *RRM2,* which remained just 2.8-fold higher, while *BIRC5* expression returned to pre-infection levels. Together, these findings support a model wherein MAIT cell activation with increased cell cycling occurs in the earliest stages of acute HIV-1 infection, and then subsides as disease progresses into chronic infection.

**Upregulation of innate immune pathways at peak viremia.** To examine the MAIT cell transcriptome at the pathway level, gene set enrichment analysis (GSEA) at the pre-infection and post-infection time points was performed[42,43]. GSEA analysis using the Gene Ontology (GO) gene set revealed an enrichment of multiple pathways at one or several time points during acute HIV-1 infection (Fig. 3d and Supplementary Table 5). Many enriched gene sets were related to cellular activation and metabolism, DNA replication, or cell cycle progression, in line with the observed patterns of MAIT cell activation and expansion. However, several important immunological pathways were also upregulated, including the gene signatures for negative regulation of viral entry (Fig. 3e), positive regulation of IFNγ production (Fig. 3f), and natural killer (NK) cell mediated immunity

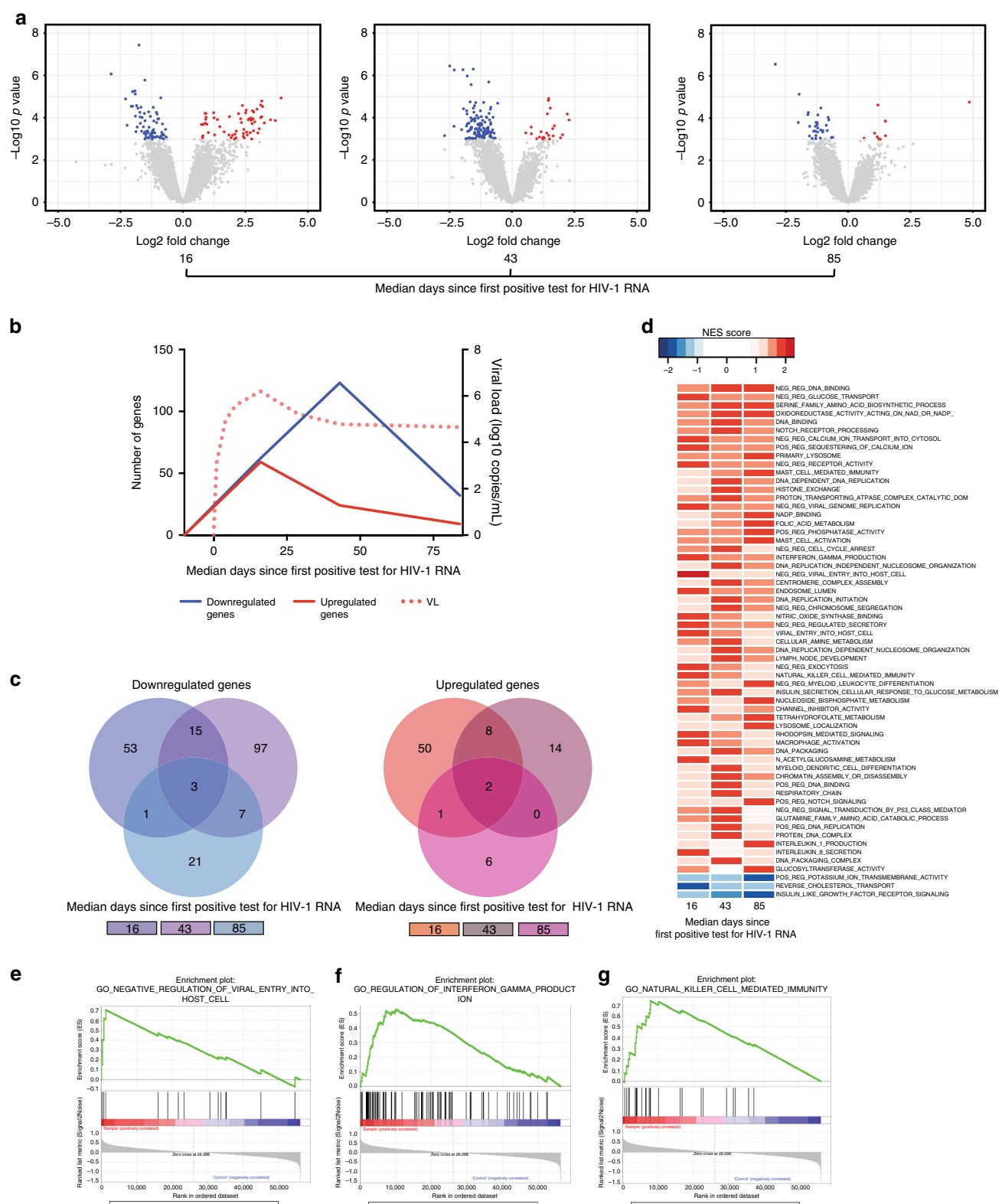

(Fig. 3g). The NK cell gene signature included enhanced expression of *KLRD1*, *GZMB*, *MICB*, *SLAMF7*, and *KIR3DL1*; the IFNγ production signature included *IFNaR1*, *TLR3*, and *HAVCR2*; and the negative regulation of viral entry pathway included upregulation of *IFITIM1, 2, and 3* and *TRIM5*. The MAIT cell GSEA data are indicative of not only activation and cell proliferation but also utilization of pathways more commonly affiliated with the innate cellular compartment.

**MAIT cell TCR repertoire diversification in acute infection.** The MR1-restricted TCR repertoire of MAIT cells is limited, in particular for the TCR α chain while the β chain shows greater diversity. Nevertheless, controlled infections of humans with *S. enterica* was recently shown to lead to preferential expansion of the more antigen reactive MAIT cell clonotypes[44]. To evaluate possible alterations in the TCR repertoire of MAIT cells resulting from acute HIV-1 infection, we analyzed the TCR α

**Fig. 3 The transcriptional signature of MAIT cells before and during acute HIV-1 infection.** RNA-Seq was performed on sorted MAIT cells from the PBMC of longitudinal samples corresponding to one pre-infection and three post-infection time points in the acute capture cohort ($n = 9$). **a** Volcano plots depict upregulated (red) or downregulated (blue) genes compared to pre-infection at three post-infection time points in acute HIV-1 infection. Individual genes listed in Supplementary Table 4. Highlighted genes have a –$\log_{10}$ $p$-value $\geq 3$ and a $\log_2$ fold change of 0.5 or −0.5 (corresponding to $p \leq 0.001$, and fold change of 1 or −1, in a generalized linear model). **b** The temporal dynamics of the upregulated and downregulated genes shown longitudinally in acute HIV-1 infection, together with plasma VL. **c** Shared and unshared differently expressed genes compared to pre-infection between all three post-infection time points in acute HIV infection are highlighted as a Venn diagram, and listed in Supplementary Table 4. **d** Gene expression patterns were subjected to Gene Set Enrichment Analysis (GSEA), and upregulated and downregulated pathways in post-infection time points compared to pre-infection are displayed as a Normalized Enrichment Score (NES) heat map. Enrichment plots from three selected post-infection upregulated pathways compared to pre-infection are shown; **e**, negative regulation of viral entry into host cell, **f**, regulation of IFNγ production, and **g**, natural killer cell mediated immunity. Genes contributing to enrichment plots are listed in Supplementary Table 5. PBMC, Peripheral blood mononuclear cells. VL, viral load. MAIT cells are identified as CD161 + + Vα7.2 + cells within CD3 + CD14-CD19- live lymphocytes.

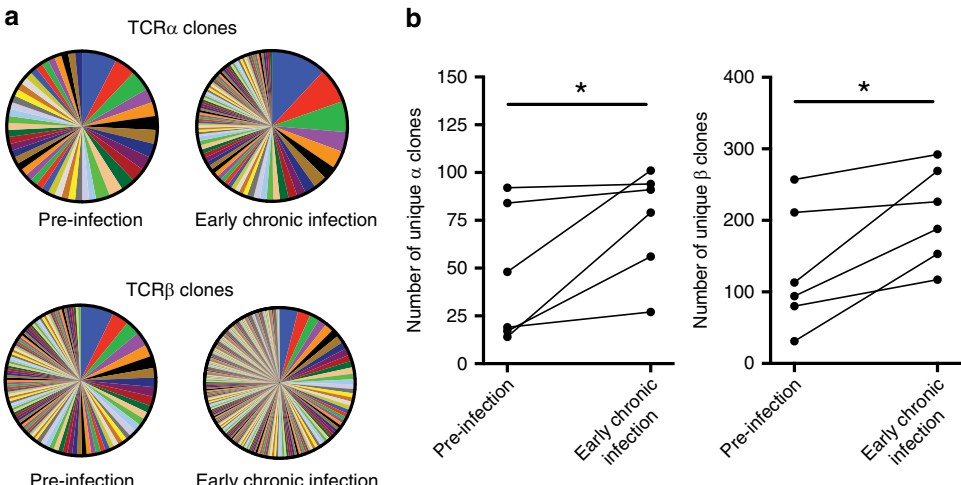

**Fig. 4 TCR repertoire diversity of MAIT cells before and after acute HIV-1 infection.** RNA-Seq was performed on sorted MAIT cells from PBMC samples at pre-infection and early chronic post-infection time points in the acute capture cohort ($n = 6$). **a** TCRα and TCRβ clonal sequence diversity of MAIT cells before and after HIV-1 infection in one representative donor. **b** Change in the number of MAIT cell TCRα and TCRβ clones detected before and after HIV-1 infection. *$p < 0.05$, Wilcoxon Signed Rank test. MAIT cells are identified as CD161++Vα7.2 + cells within CD3+CD14-CD19- live lymphocytes. Donors analyzed here are a subset of donors analyzed in Fig. 2. Post-infection time point corresponds with early chronic HIV infection (85 days since first positive test for HIV-1 RNA).

and β chain transcripts within the RNA-Seq data at pre-infection and the early chronic time point for six donors (Fig. 4). Surprisingly, acute HIV-1 infection was associated with enhanced diversity of the CDR3 clonal distribution of both TCR α and β chains (Fig. 4a and Fig. 4b). Unique clone frequencies ranged between 0.4–22.2% (median = 6.9%) of all TCR α chain clones or 0.3–40.3% (median = 3.6%) of all β chain clones. Interestingly, two CDR3 α chain clones were shared and considered "public" among all donors and time points analyzed. The diversifying effect of acute HIV-1 infection was also associated with the contraction of the dominant α and β chain clones pre-infection, and specifically of TRBV20−1 (Vβ2) usage, in the MAIT cell repertoire (Supplementary Fig. 4 and Supplementary Table 6). These changes in TCR repertoire composition during acute infection may suggest that activation and response of the MAIT cell compartment depend on TCR-mediated recognition of microbes.

**Progressive upregulation of innate MAIT cell characteristics.** MAIT cells that express the neural-cell adhesion marker CD56, commonly associated with NK cells, have an enhanced responsiveness to the innate cytokines IL-12 and IL-18 in healthy humans[23]. Given the results of the GSEA pathway analysis, the behavior of the CD56+ MAIT cell subset during acute HIV-1 infection was evaluated (Fig. 5a). Interestingly, the proportion of

CD56+ MAIT cells among the total MAIT cell population increased progressively throughout acute infection from an average of 32% before infection to an average of 47% of total MAIT cells in chronic infection (Fig. 5b). This pattern was also reflected in the CD56 real-time PCR gene expression data from sorted total MAIT cells (Fig. 5c). Furthermore, several additional transcripts associated with innate effector cell function were upregulated throughout acute infection, compared to pre-infection, including NKG7, KLRD1, EOMES, CD160, SLAMF5, and IL12RB1 (Fig. 5d).

To assess whether the expanding CD56+ MAIT cell subset had superior capacity to respond to innate cytokine stimulation[23], PBMC were stimulated with IL-12 and IL-18, and IFNγ production in subsets of MAIT cells with or without CD56 surface expression was evaluated by intracellular cytokine staining and flow cytometry (Fig. 5e). In samples drawn before HIV-1 infection, MAIT cell responses to the cytokine stimulus recapitulated the pattern previously reported for healthy donors, with higher IFNγ expression in the CD56+ MAIT cells compared to their CD56− counterparts ($p < 0.01$; Wilcoxon Signed Rank test) (Fig. 5e, f and Supplementary Fig. 5). Importantly, this pattern was retained post-infection, with no sign of decline in IL-12 and IL-18 responsiveness during early chronic HIV-1 infection (Supplementary Fig. 5). Together, the expansion of CD56+ MAIT cells represents a progressive increase of innate characteristics within the MAIT cell compartment throughout acute HIV-1 infection.

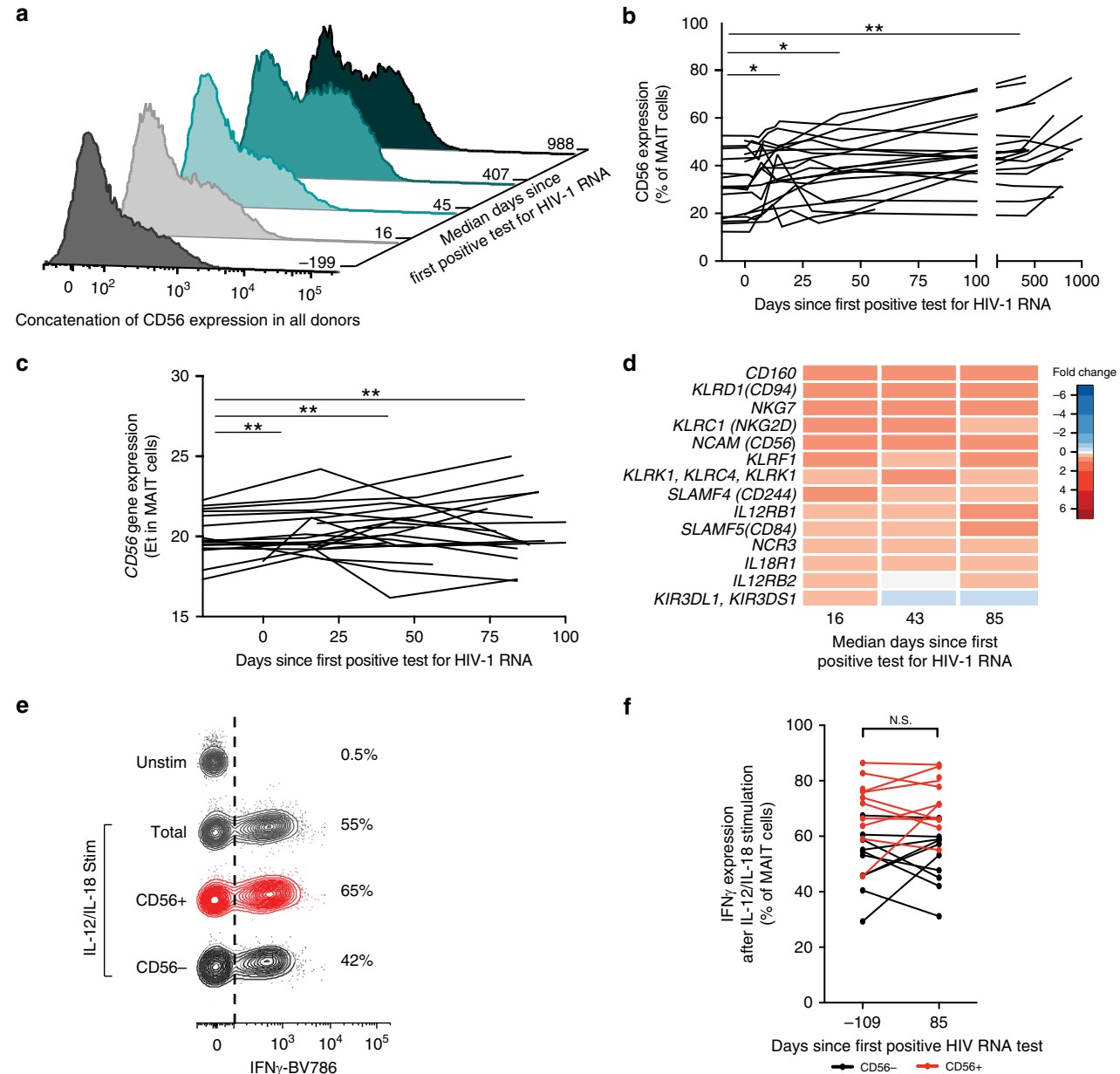

**Fig. 5 MAIT cells display increased innate-like properties in acute HIV infection. a** Flow cytometry was performed on PBMC from individuals in the acute capture cohort to investigate changes in CD56 expression within MAIT cells, and concatenation of all individuals examined are shown over time in acute HIV-1 infection (n = 19). **b** CD56 expression in MAIT cells is displayed longitudinally at the protein level measured by flow cytometry, and **c**, at the gene expression level as measured by targeted transcriptomics (Et = #of qPCR cycles−Ct) (n = 19). **d** The fold change compared to pre-infection of innate-like genes after targeted transcriptomics of bulk sorted MAIT cells at three post-infection time points in acute infection (n = 20). **e** Example flow cytometry staining of IFNγ production in CD56− or CD56+ MAIT cells from a pre-infection time point from one donor in the acute capture cohort after stimulation of PBMC with IL-12 and IL-18 (n = 10). **f** IFNγ production in CD56 + and CD56− MAIT cells after stimulation of PBMC with IL-12 and IL-18 at one pre-infection and one post-infection time point (n = 10). *p ≤ 0.05, **p ≤ 0.01. In **b** and **c**, statistical analysis was performed using the nonparametric Friedman test with the Dunn's multiple comparison test; in **f**, using Wilcoxon Signed Rank test. PBMC, Peripheral blood mononuclear cells. MAIT cells are identified as CD161 + + Vα7.2 + cells within CD3 + CD14-CD19- live lymphocytes. Post-infection time point corresponds with early chronic HIV infection (85 days since first positive test for HIV-1 RNA). The source data underlying **d** are provided as a Source Data file.

**Transiently elevated MAIT cell function precedes decline.** We next investigated MAIT cell functionality in response to bacterial or mitogen stimulus. MAIT cell responses in PBMC pulsed with *E. coli*, or in response to PMA/Ionomycin, were detected by intracellular cytokine expression as previously described[45] (Fig. 6a). Compared to pre-infection, MAIT cell expression of single functions did not change in response to *E. coli*, as tendencies towards increased expression of GrzB at the time of viral load set-point did not reach significance (Fig. 6b). MAIT cell

production of cytokines in response to bacterial stimulation were rather stable during the first three time points evaluated, and showed a trend towards decline at the final time point corresponding to early chronic HIV-1 infection (Fig. 6b). This pattern was also observed in PMA/ionomycin stimulation, with reduced TNF expression at the final time point tested (p = 0.036; Friedman test with the Dunn's multiple comparison test) (Fig. 6c). Interestingly, determination of the total functionality in MAIT cells in response to *E. coli*, calculated as the percentage of

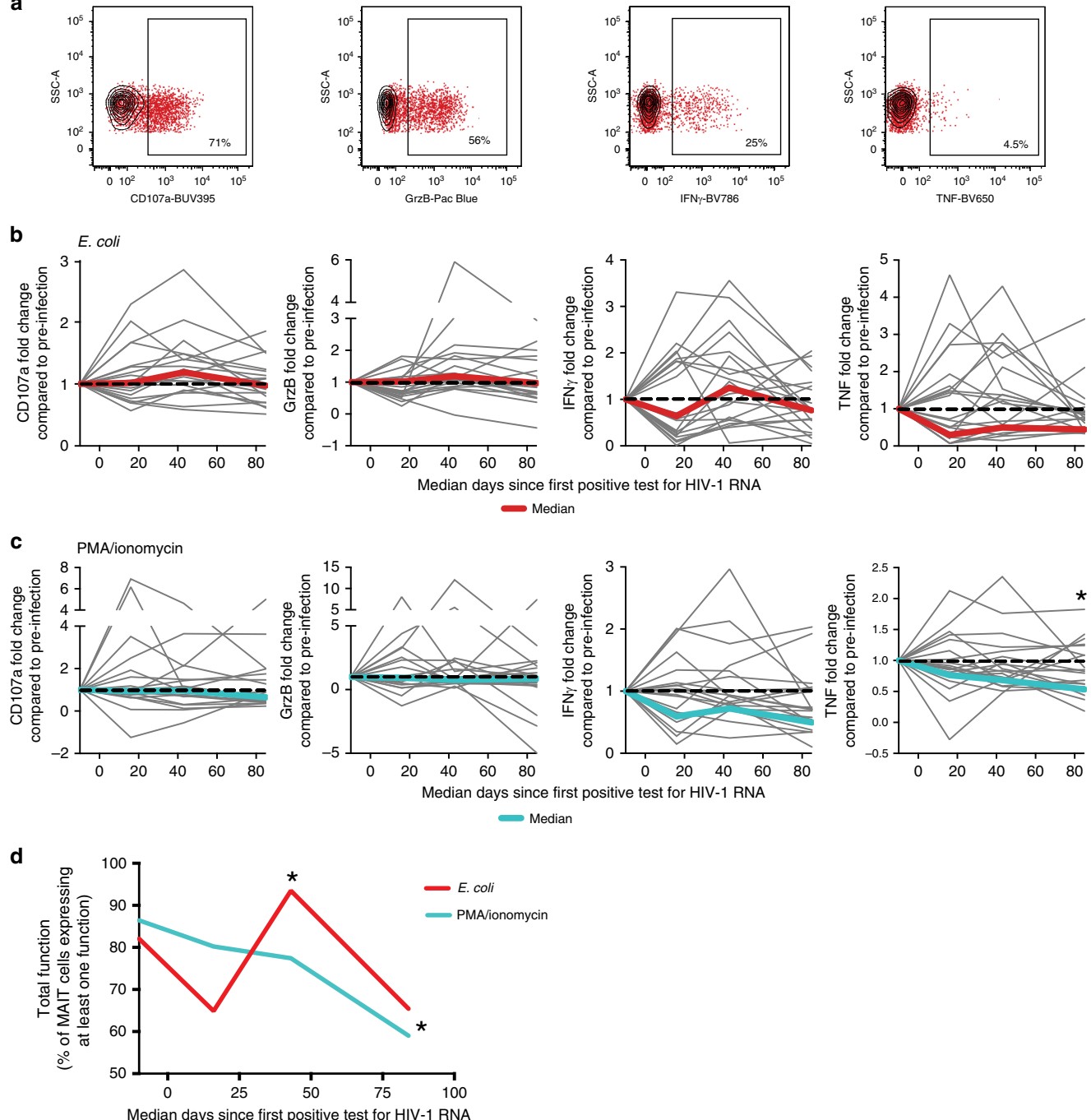

**Fig. 6 MAIT cells in acute HIV-1 infection become dysfunctional in early chronic infection.** PBMC from one pre-infection and three post-infection time points from the acute capture cohort study subjects ($n = 20$) were stimulated with *E. coli*, PMA/ionomycin, or without stimulation, to examine expression of markers of cytotoxicity (CD107a and GrzB) and cytokine production (IFNγ and TNF) in MAIT cells. **a** Example flow cytometry gating showing the functional read out within the MAIT cell gate in unstimulated (black) and stimulated (red) cells. **b** MAIT cell functionality after stimulation with mildly fixed *E. coli*, or **c** PMA/ionomycin is displayed longitudinally with data from one pre-infection and three post-infection time points in acute infection as the percentage of MAIT cells positive for functional markers, and the median of these markers at pre-infection is shown dashed black line. Longitudinal median values shown as a solid red or turquoise line, respectively. **d** The percentage of MAIT cells from **b** and **c** expressing at least one function is shown longitudinally. *$p \leq 0.05$. In **b**, **c**, and **d**, statistical analysis was performed using the nonparametric Friedman test with the Dunn's multiple comparison test. PBMC, Peripheral blood mononuclear cells. MAIT cells are identified as CD161++Vα7.2+ cells within CD3+CD14-CD19- live lymphocytes. VL viral load. The source data underlying **d** are provided as a Source Data file.

MAIT cells expressing at least one function in response to stimulus (Fig. 6d), showed a pattern where this measure of MAIT cell responses was transiently enhanced at the time of viral load set-point ($p = 0.04$; Friedman test with the Dunn's multiple comparison test). This pattern reversed into a decline in functionality at the early chronic time point in PMA/ionomycin stimulated MAIT cells, as compared to baseline ($p = 0.01$; Friedman test with the Dunn's multiple comparison test).

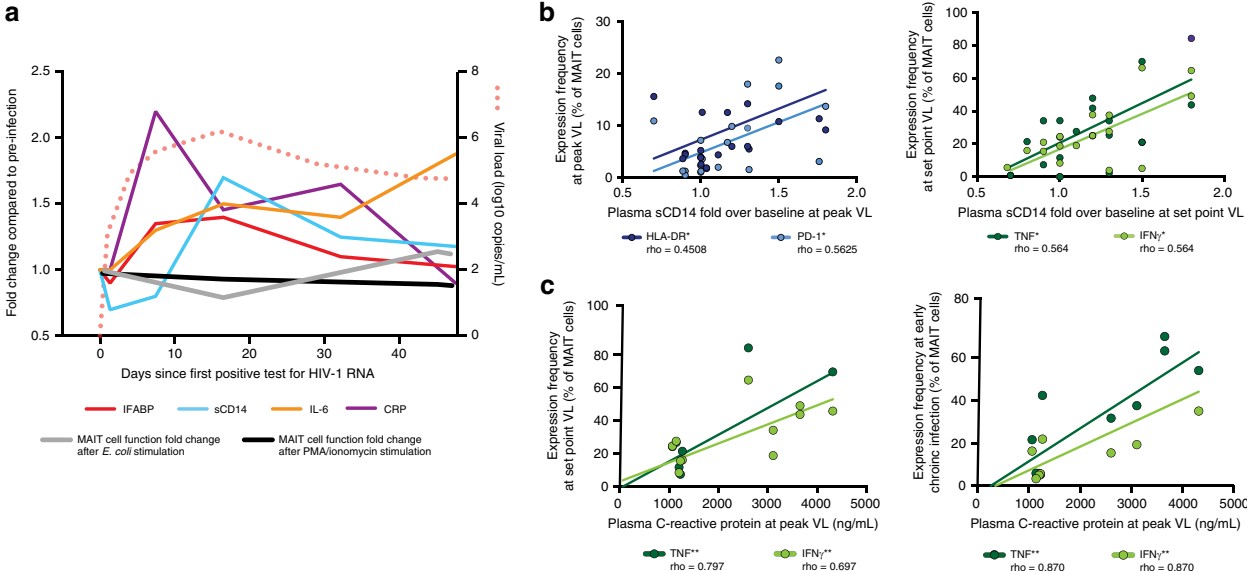

**Fig. 7 Plasma soluble factor associations with MAIT cell phenotype and function in acute HIV-1 infection. a** The fold change compared to pre-infection of plasma soluble factors (IFABP, sCD14, IL-6, and CRP) and total MAIT cell function after *E. coli* or PMA/ionomycin stimulation from individuals enrolled in the acute capture cohort, displayed longitudinally (*n* = 20). **b** Correlative analysis of plasma sCD14 fold over baseline at peak VL (median 16 days since first positive test for HIV-1 RNA) or set point VL (median 43 days since first positive test for HIV-1 RNA) with either the expression of activation markers (HLA-DR and PD-1) on MAIT cells, or the production of TNF and IFNγ in MAIT cells after PMA/ionomycin stimulation, respectively (*n* = 20). **c** Predictive correlation of plasma CRP at the peak VL time point with MAIT cell functionality after PMA/ionomycin stimulation at either the set point VL time point (median 43 days since first positive test for HIV-1 RNA), or the early chronic infection time point (median 85 days since first positive test for HIV-1 RNA) (*n* = 10). \*\**p* ≤ 0.01, \*\*\**p* ≤ 0.001. In **b** and **c**, correlative analyses were performed using Spearman Rank correlation test. PBMC, Peripheral blood mononuclear cells. MAIT cells are identified as CD161++Vα7.2+ cells within CD3+CD14-CD19- live lymphocytes. VL viral load. IFABP intestinal fatty acid binding protein. sCD14 soluble CD14. CRP C-reactive protein. The source data underlying **a** are provided as a Source Data file.

**MAIT cell responses are associated with sCD14 and CRP levels**. Acute HIV-1 infection is associated with massive release of cytokines including mediators of inflammation and a rise in markers of microbial translocation[46]. Here we assessed plasma levels of soluble CD14 (sCD14), C-reactive protein (CRP), IL-6, and intestinal fatty acid binding protein (IFABP) at the pre-infection time point and over the course of the first 45 days of acute HIV-1 infection (Fig. 7a). Levels of CRP spiked very early in infection and returned to normal by day 45, whereas the rise in IFABP and sCD14 occurred with slower kinetics and did not fully normalize to levels observed prior to infection. In contrast, IL-6 concentrations continued to rise over the course of this early stage of infection. Interestingly, levels of sCD14, a marker of monocyte activation and microbial translocation, correlated directly with concurrent MAIT cell activation at peak viremia, and were positively associated with MAIT cell production of TNF and IFNγ at the time of set-point viral load (Fig. 7b). Also, levels of CRP at peak viremia correlated with TNF and IFNγ production by MAIT cells at both set-point viral load and early chronic infection time point (Fig. 7c). Thus, microbial translocation and acute-phase inflammatory responses may influence MAIT cell activation and function during acute HIV-1 infection.

## Discussion

Unconventional T cell subsets, such as MAIT cells, that recognize non-peptide antigens presented by MHC class I-like molecules broaden the scope of microbial antigen recognition beyond proteins[47]. Many microbes of clinical relevance in the immuno-compromised host, such as mycobacteria, express the riboflavin pathway and can thus give rise to the main class of MR1-presented antigens recognized by MAIT cells. Here, we investigate the response of MAIT cells during the first critical days and

weeks of acute HIV-1 infection utilizing longitudinal pre-infection and post-infection samples from the RV217 ECHO study[34], and show that the MAIT cell compartment responds rapidly during acute HIV-1 infection (Fig. 8). At peak viremia MAIT cells are already activated with elevated expression of CD38, HLA-DR, TIGIT, and PD-1, and show clear signs of transcriptional activation with upregulation of transcripts involved in cell cycle progression and cell division. Interestingly, MAIT cell activation correlates directly with plasma levels of sCD14 at peak viremia. Even though sCD14 may not exclusively mark microbial translocation, this correlation supports the model where MAIT cell activation is driven by exposure to microbial products and microbial antigens. Notably, the activation of MAIT cells at peak viremia occurs at a stage in infection when depletion of innate lymphoid cells occurs concomitant with signs of gut barrier breakdown[48]. Furthermore, at peak viremia MAIT cell levels of activation markers correlate directly with levels of transcripts for *KI67* and *IRF4*. This pattern is consistent with initiation of proliferation and generation of effector-type MAIT cells at this early phase of acute infection.

It is well established that MAIT cells suffer numerical and functional decline in chronic untreated HIV-1 infection, but the timing of this loss is unknown. Here, we demonstrate that the initial signs of numerical loss occur beyond one year into HIV-1 infection, whereas no loss is seen during acute stages. In fact, the initial activation is followed by significant expansion of the MAIT cell compartment just after peak viremia, occurring both in peripheral blood and in rectal mucosa. This is accompanied by an enhanced response against *E. coli*-pulsed cells in terms of total functionality, i.e., the ability of MAIT cells to respond with any measured function. It thus seems like the initial wave of activation at peak viremia causes a priming phenomenon, which is followed by an effector response with expanded levels of

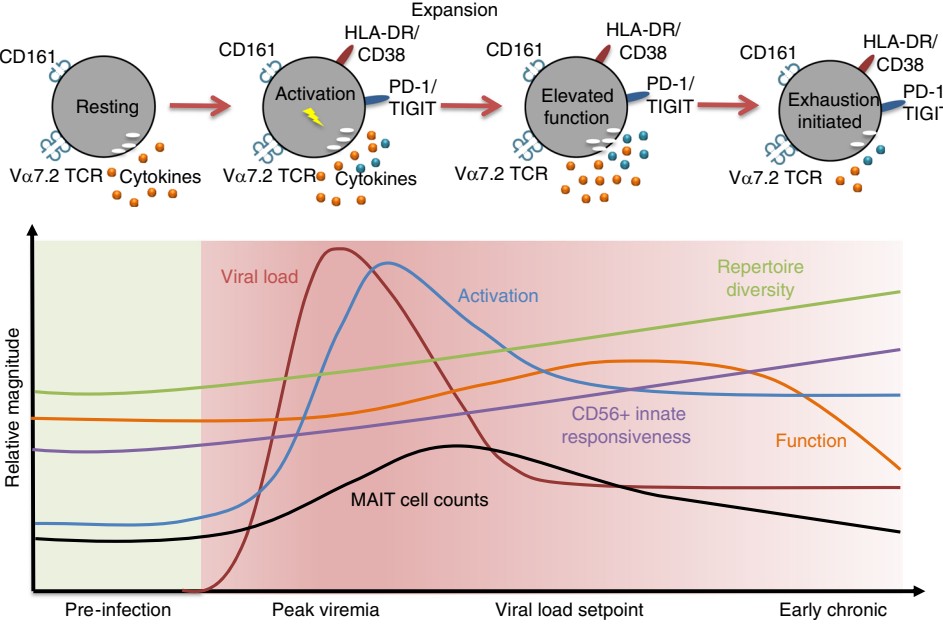

**Fig. 8 Dynamic response of the MAIT cells compartment during acute HIV-1 infection.** Schematic view of the complex response pattern of MAIT cells from pre-infection, to peak viremia, over the viral load set-point, into early chronic infection.

effector MAIT cells. Similar to the correlation between activation markers and sCD14 at peak viremia, the functional capacity at the time of set-point viral load correlates directly with sCD14. It is thus possible that this response is driven by microbial translocation and direct TCR engagement. Consistent with this, the TCR repertoire of the MAIT cell compartment changes in response to acute HIV-1 infection, with enhanced TCR diversity as infection progresses into early chronicity. This enhancement of diversity seems to be driven by expansion of some clones, decline of some and addition of new clones not detected at the pre-infection time point. Controlled infections of humans with *S. enterica* was recently shown to lead to preferential expansion of the more antigen reactive MAIT cell clonotypes[44]. The more mixed response pattern observed here during acute HIV-1 infection is consistent with a broad stimulus by diverse microbes via microbial translocation.

The initial activation and expansion of MAIT cells during acute HIV-1 infection is eventually followed by decreased activation, as well as numerical and functional loss. Interestingly, expression of the transcription factor *IRF4* at peak viremia, a gene whose induction and sustained expression in conventional CD8 T cells corresponds to exhaustion in viral infection[49], correlates both directly with concurrent PD-1 expression levels, and inversely with MAIT cell counts and percentages at later time points. This opens the possibility that strong activation at very early stages during infection may set the MAIT cell compartment up for exhaustion at later stages. Another interesting aspect is that functional decline seems to precede MAIT cell loss. It is possible that these two facets of the MAIT cell compartment are independent from each other. For example, progressive homing of functional cells to peripheral sites may leave the less functional or exhausted MAIT cells in circulation. This possibility is supported by the increase in CCR5 expression by MAIT cells in early chronic stages of infection in the absence of evidence of caspase activation or caspase transcript upregulation, and also by the increased levels of MAIT cells detected in rectal mucosa. Thus, homing of MAIT cells to sites of inflammation and microbial translocation remains a primary candidate explanation for the loss of MAIT cells in circulation[31].

The transcriptional landscape in MAIT cells during acute HIV-1 infection shows activation of a range of pathways, including those involved in cell cycle progression and proliferation. Several important immunological pathways were also upregulated, including a gene signature for NK cell-mediated immunity. Furthermore, there is progressively enhanced frequency of CD56 expression in the MAIT cell compartment over time throughout acute and into early chronic infection. This is notable because CD56 marks a MAIT cell subset with enhanced responsiveness to innate cytokine stimulation in healthy donors[23], a pattern confirmed here for HIV-1 infected individuals. It is important to note that the MAIT cell responsiveness to IL-12 and IL-18 stimulation is preserved in matched donor samples before and after HIV-1 infection. Overall, these data are consistent with progressively enhanced innateness in the MAIT cell compartment in response to HIV-1 infection, suggesting that MAIT cells move along the recently described innate-adaptive spectrum of unconventional T cells[50].

The MAIT cell pool is mostly CD8+, with smaller DN and CD4+ subsets. The longitudinal pattern over acute HIV-1 infection reveals changes in the subset composition as compared to the pre-infection time point, with relatively fewer CD8+ and CD4+ MAIT cells, and an expanding DN MAIT cell subset. The shift from CD8+ to DN character is consistent with the expansion of effector-like MAIT cells, and with recent observations indicating that the DN subset is an activation-induced derivative of the main CD8+ MAIT cell pool[51]. CD8+ MAIT cells give rise to DN MAIT cells as part of their effector response, and this may be important given that the DN population is functionally distinct with enhanced IL-17 and lower IFNγ production[51]. The decline in CD4+ MAIT cells observed here is different from the preservation of CD4+ MAIT cells seen in previous studies of chronic HIV-1 infection, and may indicate that this small subpopulation is susceptible to infection. The difference between the present data and previously published data may be explained by the fact that CD4+Vα7.2+CD161+ cells include many contaminating non-MAIT cells and only approximately 30–50% of this population is bona fide MAIT cells[23]. The present dataset is based on MR1-tetramer defined populations, which excludes such non-MAIT cells from analysis.

The previous observation by us and others that MAIT cells are lost in chronic HIV-1 infection and generally do not recover in response to successful cART[24,25], indicated a persistent impairment of this broad anti-microbial defense mechanism. In this context, it is significant that this study indicates that the numerical loss of MAIT cells does not occur within the first year of infection. Some signs of functional impairment are evident already at three months post infection but this may not be as critical, since previous studies suggest that antibacterial responsiveness can at least be partly restored in residual MAIT cells upon cART later in chronic infection[24]. Overall, this pattern indicates that the MAIT cell compartment is not subject to rapid degradation in the first months of HIV-1 infection, and can probably be rescued by cART within the first months after infection. There is thus a fairly generous window of opportunity to preserve this important arm of anti-microbial immunity by initiation of treatment within this period, but this ultimately needs to be explored to better understand how early therapy can ameliorate immune dysfunction.

It is interesting to note similarities between our observations in acute HIV infection, and those made by others studying SIV infection in non-human primates. Juno et al. recently observed activation and expansion of MAIT cells in pigtail macaques acutely infected with various SIV strains, with little evidence of MAIT cell loss during the first nine months of infection[52]. In chronically SIV infected Asian macaques, Vinton et al. observed activation, proliferation and loss of MAIT cells[53]. Whereas differences in host species and viral strain characteristics are important, these patterns are broadly consistent with the patterns observed in HIV-1 infected humans.

In summary, these findings delineate the dynamic MAIT cell response to acute HIV-1 infection where the MAIT cell compartment initially responds and expands with enhanced function, goes through TCR repertoire diversification, followed by progressive reprogramming away from TCR-dependent antibacterial responses and towards enhanced innateness. These changes are likely to have broad consequences for antimicrobial immunity in humans infected with HIV-1.

## Methods

**Ethics.** All subjects in studies RV217/WRAIR#1373, RV254/SEARCH 010/WRAIR#1494, and RV304/SEARCH 013/WRAIR#1751 were adults and provided written informed consent. For subjects that were unable to read, the consent document was read to them with an impartial witness present; the volunteer, the witness and the study staff obtaining consent signed the affidavit with a signature or mark. Studies were reviewed and approved by the human subject ethics and safety committees in each country, as well as by the Walter Reed Army Institute of Research (WRAIR) (Silver Spring, MD, USA), in compliance with relevant federal guidelines and institutional policies. RV304: The Institutional Review Board of the Faculty of Medicine, Chulalongkorn University; and the WRAIR Institutional Review Board. RV217: Institutional Review Board Royal Thai Army Medical Department; Kenya Medical Research Institute (KEMRI) Scientific and Ethics Review Unit (SERU); Uganda National HIV/AIDS Research Committee (NARC); Mbeya Medical Research and Ethics Committee (MMREC) and the National Health Research Ethics Committee (NatHREC); and the WRAIR Institutional Review Board. RV254: The Institutional Review Board of the Faculty of Medicine, Chulalongkorn University; and the WRAIR Institutional Review Board.

**Subjects.** This study focuses on 29 participants from the RV217 ECHO cohort (Supplementary Table 1)[34]. The RV217 study enrolled high-risk, consenting adults at four clinical research sites: Walter Reed Project, Kericho, Kenya; Makerere University Walter Reed Project, Kampala, Uganda; Mbeya Medical Research Center, Mbeya, Tanzania; and Armed Forces Research Institute of Medical Sciences, Bangkok, Thailand. HIV-uninfected participants were screened twice weekly with small samples through finger pricks, and analyzed with nucleic acid amplification test (Aptima HIV-1 RNA Qualitative test, Hologic Inc., San Diego, CA). Enrollees with reactive tests were enrolled in a second phase of the study that included twice-weekly sampling of large blood volumes for one month. Upon HIV-1 confirmation by standard serological methods, HIV acute cases were offered participation in long-term follow up phase. All HIV-1 positive participants were referred to care providers for management of the infection, based on national

guidelines. The cases presented in this study are a selected set from a group of 29 RV217 participants for which cryopreserved PBMC were available at pre-infection, and at least three post-infection time points corresponding to peak viral load (median 16 days, range 14–22, since first positive test for HIV-1 RNA), set point viral load (median 43 days, range 31–50, since first positive test for HIV-1 RNA), and early chronic infection (median 85 days, range 60–126, since first positive test for HIV-1 RNA). Occasionally, long term follow up samples were used (out to 1,040 days since first positive test for HIV-1 RNA).

Additional cross sectional studies were performed with samples from 7 acutely HIV-1 infected participants from the RV254/SEARCH 010 study sampled before initiation of cART, and 17 uninfected matched controls from the RV304/SEARCH 013 study[39] (Supplementary Table 2).

**Clinical parameters.** Plasma HIV-1 RNA levels were measured using the Real-Time HIV-1 Assay (m2000 RealTime System, Abbott Molecular). EDTA-anticoagulated samples of whole blood were analyzed with the use of the BD Multitest on a FACSCalibur flow cytometer (BD Biosciences).

**Targeted quantitative gene expression analysis.** CD161++Vα7.2+MAIT cells from 20 individuals from the RV217 acute capture cohort from one pre-infection time point and three post-infection time points were sorted (100 cells/well) directly into reaction mixture (SuperScript III Reverse Transcriptase/Platinum Taq Mix, Cells Direct 2× Reaction Mix, Invitrogen). Reverse transcription and specific transcript amplification were performed using a thermocycler (Applied Biosystems Gene Amp PCR System 9700) with the following parameters: 50 °C for 15 min, 95 °C for 2 min, and 95 °C for 15 s; and then 60 °C for 30 s for 18 cycles. Amplified cDNA was then loaded into a Biomark 96.96 Dynamic Array chip using the Nanoflex IFC controller (Fluidigm). This microfluidic platform was then used to conduct quantitative qPCR (qPCR). Threshold cycle (CT), as a measurement of relative fluorescence intensity, was extracted from the BioMark Real-Time PCR Analysis software. See Supplemental Experimental Procedures and Supplementary Table 7 for details.

**RNA-Seq transcriptomics.** Peripheral blood MAIT cells were purified by sorting (1911–64,011 total cells) using a FACS Aria SORP (BD Biosciences), pelleted, and overlaid with 250 μl of RNAlater (ThermoFisher) and frozen at −20 °C. RNA was extracted using the RNeasy Mini Kit (Qiagen), and RNA quality and concentration were assessed with the Agilent 2100 Bioanalyzer Pico Chip. RNA-Seq libraries were prepared using the SMART-Seq v4 Ultra Low Input RNA Kit (Clontech) according to the manufacturer's instructions. Amplified material was purified using Agencourt AMPure XP beads (Beckman). cDNA quantity was assessed on a Qubit 3.0 (ThermoFisher) and fragment size was evaluated on a 2100 BioAnalyzer (Agilent). The PCR products were next indexed using the Nextera XT DNA Library Prep Kit (Illumina) according to the manufacturer's instructions. Briefly, products were tagmented using the Amplicon tagment mix containing Tn5 transposase, and indexed using Nextera index 1 (i7) and index 2 (i5) primers. The libraries were again cleaned-up with Agencourt AMPure XP beads, quantified, pooled, and sequenced across 75 base pairs (bp) using a single-end strategy with a 75-cycle high output flow cell on a NextSeq 500 (Illumina). Nine biological replicates were sequenced per experiment, with four donor-matched time points corresponding to one pre-infection and three post HIV-infection time points at peak viral load, set point viral load, and early chronic infection. Median reads per sample was 22.9 million reads. The Unix based program Spliced Transcripts Alignment to a Reference (STAR) v.2.6.1 with human genome hg38, was used for alignment[54]. Transcription mapping was performed using RNA-seq by Expectation Maximization (RSEM) v.1.3.1[55]. The feature Counts program was used for counting mapped reads[56]. RUVSeq v1.12 was used to remove unwanted variation[57], and differentially expressed gene list was generated by edgeR v3.20. R package[58]. The GSEA method[42] was used for finding statistically significant pathways with 5917 gene sets of GO in Molecular Signatures Database (MSigDB) issued by Broad Institute. TCR data was extracted from the RNA-Seq dataset from six donors with sufficiently high cell counts across all four analyzed time points, using the MiXCR software (https://mixcr.readthedocs.io/en/master/index.html). See Supplemental Experimental Procedures for details on TCR analysis.

**Polychromatic flow cytometry.** Four polychromatic flow cytometry panels were used to measure MAIT cell function, phenotype, and for cell sorting for transcriptomics[59]. Briefly, thawed samples were washed, stained with LIVE/DEAD Fixable Aqua Dead Cell dye (ThermoFisher), blocked for Fc receptors using Normal mouse serum (ThermoFisher), and surface stained with antibody cocktail. Samples were surface stained at room temperature for 30 min, and some were intracellularly stained at room temperature for 30 min. Some samples were fixed in 2% paraformaldehyde or BD FIX/PERM Buffer (BD Biosciences) before acquisition on a 5 laser, 16-parameter BD LSRII SORP flow cytometer (BD Biosciences) within 12 h of staining. Other samples used for sorting for downstream transcriptomics were resuspended in sorting buffer (PBS containing 1% BSA) and sorted for bulk MAIT cells for either RNA-Seq or targeted transcriptomics with Fluidigm Biomark. Data were analyzed with FlowJo v.9.9.4 (TreeStar). See Supplemental Experimental

Procedures and Supplementary Table 8 for specific antibodies, MR1 tetramer, and reagents used in the study.

**MAIT cell functional assay**. MAIT cell functionality pre-HIV and post-HIV infection was assessed using three stimulation techniques. Detailed experimental procedures and specific antibodies used are described in Supplemental Experimental materials. Briefly, PBMC were stimulated for either 24 h with partially fixed *E. coli* D21 in the presence of anti-CD28, for 24 h with IL-12 and IL-18, or for 6 h with PMA/ionomycin as per the manufacturer's recommendation (eBioscience™ Cell Stimulation Cocktail (500×), ThermoFisher). All stimulation methods included BFA and monensin for the last 6 h of stimulation.

**Soluble cytokine analysis**. Luminex® based detection assays were used to measure plasma levels of C-reactive protein (CRP) and IL-6 (EMD Millipore, Billerica MA) per manufacturer's instructions. Briefly, samples were mixed with a cocktail of MagPlex® magnetic microspheres, bound to capture antibody specific to proteins of interest. Following incubation with sample overnight at 4 °C, excess sample was washed off using a magnetic plate washer (BioTek Instruments, Winooski VT) and biotinylated detection antibody cocktail was added for 1 h at room temperature. Streptavidin-phycoerythrin was added for 30 min before a final wash and resuspension in sheath fluid. Data was collected on a FlexMap 3D® system. Levels of sCD14 and intestinal fatty acid binding protein (IFABP) were measured by standard chemiluminescent detection ELISA (R&D Systems, Minneapolis MN) per manufacturer's instructions and read on a VersaMax® reader (Molecular Devices, Sunnyvale CA). Data were analyzed in Prism version 6.0 for Mac OS X (GraphPad, La Jolla CA) using a 4-parameter fit standard curve.

**Isolation of sigmoid colon mucosal mononuclear cells**. Sampling of gut-associated lymphoid tissue was performed by sigmoidoscopy, and mucosal mononuclear cells (MMCs) were isolated[39]. Briefly, 20–25 pieces of gut-associated lymphoid tissue were collected from the sigmoid colon by sigmoidoscopy using Radial Jaw 3 biopsy forceps (Boston Scientific, Natick, MA, USA). The biopsy pieces were placed in complete RPMI 1640 media containing 10% human AB serum (HAB; Gemini Bio-Product, West Sacramento, CA, USA), 1% HEPES, 1% L-Glutamine, 0.1% Gentamicin (Invitrogen, Carlsbad, CA, USA), 1% Penicillin/Streptomycin and 2.5 µg/ml Amphotericin B (Invitrogen, Carlsbad, CA, USA). Samples were then digested using 0.5 mg/ml Collagenase II (Sigma, St. Louis, MO, USA). Isolated MMCs from one donor were pooled, washed twice and then counted using Trypan Blue exclusion. MMCs were directly used for phenotypical characterization by flow cytometry.

**Statistical analyses**. The Wilcoxon Signed Rank test was used for comparison of non-parametrically distributed paired data sets. For comparison of unpaired data the Mann–Whitney test was used. The Spearman rank correlation test was used to compare correlation between two parameters. For matched longitudinal analysis the nonparametric Friedman test was performed with the Dunn's multiple comparison test. Statistical analyses were performed with GraphPad Prism v.6.0c (GraphPad Software).

**Reporting summary**. Further information on research design is available in the Nature Research Reporting Summary linked to this article.

## Data availability

The authors declare that the data supporting the findings of this study are available within the article and its supplementary information files or are available upon reasonable requests to the authors. The RNA-Seq data have been deposited in NCBI's Gene Expression Omnibus and are accessible through GEO accession number GSE126752. The source data underlying medians in Fig. 1b, c and g; 2a and b; 5d, 6d and 7a are provided as a source data file.

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

## Acknowledgements

The MR1 tetramer technology was developed jointly by Dr. James McCluskey, Dr. Jamie Rossjohn, and Dr. David Fairlie, and the material was produced by the NIH Tetramer Core Facility as permitted to be distributed by the University of Melbourne. RNA-Seq samples were processed in the Molecular Analysis Core Facility at SUNY Upstate Medical University (Syracuse, NY). The authors would like to thank the RV217 Study Team, as well as the RV304/SEARCH 013 and RV254/SEARCH 010 Study Groups for conduct of the clinical research and collection of samples. The authors also thank Rasmi Thomas for critical review of this paper. This work was supported by a cooperative agreement (W81XWH-07-2-0067) between the Henry M. Jackson Foundation for the Advancement of Military Medicine, Inc., and the U.S. Department of Defense (DOD). This research was funded, in part, by the U.S. National Institute of Allergy and Infectious Disease. This research was further supported by grants to J.K.S. from the Swedish Research Council (2016-03052) and the Swedish Cancer Society (CAN 2017/777). B.L.S., J.K.S., and M.A.E. were jointly supported by US National Institutes of Health grant (R01DK108350). The views expressed are those of the authors and should not be construed to represent the positions of the U.S. Army or the Department of Defense. The investigators have adhered to the policies for protection of human subjects as prescribed in AR 70–25.

## Author contributions

K.G.L., M.C.C., E.L., J.D., D.P.P, A.S., B.L.S., M.A.E and J.K.S. planned the studies. K.G.L., M.C., Y.P. and B.M.S. conducted experiments. K.G.L., D.K., M.C.C., A.S. and S.J.K. analyzed data. K.G.L., M.A.E. and J.K.S. interpreted the findings. N.L.M., L.A.E. and M.L.R. oversaw the RV217 cohort. C.S. and J.A. oversaw the RV254 and RV304 cohorts. L.A.E., H.K., L.M., S.N., J.K., C.S., J.A., N.L.M. and M.L.R. oversaw sample collection. K.G.L., M.A.E. and J.K.S. wrote the paper. All authors reviewed, edited, and approved the paper.

## Competing interests

The authors declare no competing interests.

## Additional information

**Peer Review Information** *Nature Communications* thanks Shelby O'Connor, James Ussher and the other, anonymous, reviewer for their contribution to the peer review of this work. Peer reviewer reports are available.

