## [Peer Review File · Nature Communications]

Reviewers' comments:

Reviewer #1 (Remarks to the Author):

Comments to authors: Here the authors have, in intricate detail, studied the frequencies, functionality, phenotype, and transcriptional profiles of MAIT T cells in a very important cohort of HIV-infected individuals which were followed longitudinally after acquisition of HIV infection. The authors find that MAIT cells are highly activated during the acute phase of HIV infection, and then become reprogrammed towards innate-like functionality. Overall the experiments are well performed, the manuscript is well written, and the conclusions are supported by the data presented.

1. The authors should include discussion of the dynamics of MAIT cells in SIV-infected Asian macaques (Vinton JV, Juno, JI). The authors should also include a short discussion of how MAIT cells seem to not be influenced by TB infection of NHPs (Kaufman Inf. Imm.).
2. Many of the observations have also been observed in other populations of CD8 T cells (PD1, HLADR, CD38 expression for example). If the authors gate on the CD8 T cells that are not MAIT cells, do expansions and activation of MAIT cells correlate with the other CD8 T cells? These data would point towards potential mechanisms underlying the observed phenomena.
3. The authors should mention the possible limitation of using Va7.2 and CD161 in capturing a few non MAIT cells and how MR1 tetramers can also be used to identify MAIT cells.
4. The discussion is quite verbose. This section could be shortened to focus on the biological significance of the observations and maybe novel therapeutic interventions which might be considered.

Reviewer #2 (Remarks to the Author):

In the manuscript "Dynamic MAIT cell response with progressively enhanced innateness during acute HIV-1 infection," Lal et. al. address important questions about the response of MAIT cells during acute HIV-1 infection. There have been numerous cross-sectional studies showing that MAIT cells are depleted in HIV+ individuals, compared to healthy controls. These studies have typically collected single time points from individuals with different phenotypes, but the HIV+ individuals have typically been infected for a long period of time. Therefore, there has not been an examination

of how HIV impacts MAIT cell phenotypes longitudinally, beginning from the time of acute HIV infection.

Lal et al take advantage of access to samples from the RV217 study. In this study, there are paired longitudinal samples of HIV+ patients that were collected before and soon after becoming HIV-infected, so they can determine the impact that the early phase of HIV infection has on MAIT cell populations in humans. This manuscript provides an important first examination of this population, filling an essential gap in our understanding of MAIT cells in human HIV infection.

As part of their study, these investigators do a really nice job of showing how the transcriptional profile of MAIT cells is highly dynamic during acute HIV infection, even though there is not depletion of MAIT cells during this time. This suggests that the ongoing pathogen response in the host is affecting MAIT cells, even if there are no apparent effect on the number and function of MAIT cells until chronic infection. It is likely that there is some intracellular reprogramming during acute infection that has long lasting impacts on the function of these cells in the host during chronic HIV.

Overall, this paper is unique and helps the the scientific community rethink the complex impact that HIV infection has on MAIT cells. Still, there are a few ways in which the data is presented that need clarification.

Concerns:

1. In figure 1d, they show that HIV+ individuals have a higher frequency of MAIT cells in the gut mucosa. This could be skewed because the individuals will have a lower percentage of CD4 T cells, and thus a higher percentage of CD8 T cells. They should report this data in terms of the percent of MAIT cells as a fraction of CD8 T cells, rather than as a fraction of total T cells. By normalizing to the percent of CD8 T cells, they will have accounted for an overall increase in CD8+ T cell frequency post-HIV infection. In the recent paper by Juno et al (J Immunol), they also observe an increased MAIT cell frequency in the rectum, but it is displayed as a percent of CD8 T cells, rather than CD3 T cells. Given some of the similar observations made during acute SIV infection in the Juno et al paper, it is surprising that the authors do not examine a4b7 expression in their samples, nor do they include mention of that paper in their discussion.

2. The TCR sequencing data is not clear. Methods explaining how this was performed are not readily apparent, except to say that it was derived from the RNA-seq data. The gene usage is also not described in the main text, even though other manuscripts (e.g. Gherardin et al) include which TRAV, TRAJ, and TRBV genes are being identified. They only show this sort of data briefly in supplemental figure 3. Furthermore, while a p-value is provided to say there is an increase in the number of unique alpha and beta clones, the statistical tests used to determine this increase is not explained. This lies

in stark contrast to the detail presented in another recent paper (Howson et al) who show that TCR diversity of MAIT cells was increased after Salmonella infection. Lal et al include the TCR data to suggest that HIV-induced microbial translocation has led to the diversification of the TCR, but the information on how this assay was performed and interpreted is incomplete. There is no expectation to repeat this assay, but it needs to be further explained and the authors need to ensure that any conclusions they make are not overstatements beyond what the data is showing.

3. The data in Figures 4e, 4f, and 4g are unclear. In 4e, they label the unstimulated and stimulated samples with different colors, but it is hard to identify these data in the plots. If distinguishing between the unstimulated and stimulated samples is important, then these plots for the different conditions should be shown separately. Further, is the data shown in figure 4E actually an example of the data presented in figure 4F? Lastly, Figure 4g shows no difference in the production of IFN γ between the CD56- and CD56+ cells before and after HIV infection, which stands in stark contrast to the comment in the results that suggests a 'progressive upregulation of innate characteristics in MAIT cells throughout acute HIV-1 infection'. Is the argument that there are more CD56+ MAIT cells after HIV infection, and that CD56+ cells are simply more innate-like? If so, that needs to be made clear.

4. The functional data presented in Figure 5 requires further explanation. In the text, they state that there is enhanced cytotoxic capacity of MAIT cells at the time of viral load set point, but the actual time point is not stated. Is this day 43? If so, how was significance determined? Was a statistical test performed on groups of samples, or did they compare the MAIT cell function of individual subjects over time? One of the major powers of this study is that they are following a longitudinal cohort. They should take advantage of this, where possible. If there is a reason they are not showing longitudinal data from the same individuals in Figure 5, then the reason for this should be described. In addition, figure 5d uses the value of 'total function' on the Y-axis, but it is not clear what 'total function' actually represents. This also needs to be clarified.

Reviewer: Shelby O'Connor

Reviewer #3 (Remarks to the Author):

Previous studies have shown that MAIT cells are depleted in HIV infection with reduced functionality. This has been evident as early in infection as 61 (39—86) weeks post infection

(reference 25). The basis for this depletion, however, is unknown. Furthermore, the dynamics during acute infection remain to be defined.

In this study, the authors have assessed changes in the MAIT cell population very early in HIV infection. Using samples from the RV217 cohort (in which high risk participants were screened twice weekly by NAT for infection), they were able to compare MAIT cell frequency, phenotype, and function at various timepoints post infection to pre-infection samples; post-infection they focused on peak viral load (median 16 days since first positive HIV-1 RNA), set point viral load (median 43 days), and early chronic infection (median 85 days). They also examined the frequency (but not absolute number or phenotype) of MAIT cells in sigmoid colon biopsies from untreated patients with acute infection from the RV254/SEARCH 010 study.

They report that:

- 1) MAIT cells in blood expand early in infection, as measured by frequency and absolute number (albeit a small change), before declining in early chronic infection.
- 2) there is an increase in double negative MAIT cells as infection progresses
- 3) the frequency of MAIT cells amongst T cells in the sigmoid colon increases in acute infection
- 4) there is increased transcription and expression of genes associated with activation, exhaustion (including IRF4), and proliferation (Ki67) in acute infection
- 5) the expression of some of these markers of activation, exhaustion, and proliferation inversely correlate with MAIT cell frequency and count
- 6) the transcriptome of MAIT cells varies at different time points after infection, with evidence of increased “innateness” with increasing time from infection
- 7) the diversity of MAIT TCR sequences increases with time from infection, suggesting antigenic stimulation and proliferation of different clones
- 8) the CD56+ MAIT cell population increases as a proportion of MAIT cells and that this subpopulation retains functionality in early chronic infection
- 9) that after an early increase (at set point viral load) in degranulation (CD107a) and granzyme B expression following stimulation, at later timepoints there is a decline in function, including cytokine production
- 10) that markers of microbial translocation and inflammation correlate with MAIT cell activation and cytokine production.

These findings are novel and increase our understanding of the fate of MAIT cells in early HIV infection. They will be of interest to those in both the HIV and MAIT cell fields. Overall, they support a model of increased activation, proliferation, and trafficking of MAIT cells to the gut early in infection, driven by microbial translocation, followed by increasing exhaustion and loss of function with time from infection. It remains to be determined whether similar changes are seen in MAIT cells in tissue as chronic infection progresses – this should be examined in future studies. Future studies should also examine the impact of early initiation of ART on MAIT cell abundance, phenotype, function, and tissue distribution.

Issues:

- 1) Statistical analysis. Multiple Wilcoxon Signed Rank tests have been used to compare median values between selected pairs when there are >2 groups. This does not allow for multiple comparisons and increases the risk of a type 1 error. Kruskal-Wallis or Friedman tests would allow for the multiple comparisons and be more appropriate.
- 2) There are marked changes in T cells in the gut in early HIV infection (loss of CD4 T cells, increased numbers of CD8 T cells), which complicates the interpretation of the increased frequency of MAIT cells amongst T cells. It is unknown whether the absolute number of MAIT cells in the gut is changed.
- 3) Method of quantitation of qPCR data. Gene expression levels are presented as E_t , which = # of PCR cycles - C_t . This does not allow for differences in PCR efficiency or sample input (although 100 cells were sorted per well). Why was relative expression to housekeeping genes not performed?
- 4) Insufficient information is provided on the bioinformatic analysis of RNASeq data.
- 5) Do inflammation and microbial translocation help preserve MAIT cell function into chronic infection? What is their association with MAIT cell abundance/frequency?
- 6) Significant correlations have been shown with MAIT cell frequency but not with abundance (eg figure 1h-j) – does this mean correlations with abundance were not significant? Could this be shown in supplementary data or as a table? Similarly, for the correlations shown in figure 6B and C, what other correlations were examined that were not significant?

Minor issues:

- 1) Page 6 – “declining frequency of CD8+ MAIT cells (... at consecutive timepoints)”. Not entirely clear what is meant by “consecutive timepoints”. Measurements appear to have been made at >3 timepoints post-infection. It is unclear what exactly has been compared.
- 2) For the 3 post-infection time points a median is provided for the number of days post first detection of HIV RNA. What is the range?

- 3) Figure 2c – Venn diagram. Text states 59 genes downregulated at peak viraemia, however there appear to be 72 on the diagram.
- 4) Figure 5d: decline in functionality at early chronic timepoint with *E. coli* is marked as significant on the figure but the text states $p=0.14$.
- 5) Page 11 – “cytokine storm” in acute HIV-1 infection. This terminology seems too strong. A cytokine storm in other settings is associated with hypotension and risk of organ dysfunction (eg CAR T cell therapy). Could an alternative term be used?
- 6) Page 15. Please provide a reference for cART improving MAIT cell antibacterial responsiveness later in chronic infection.
- 7) While included in the supplementary methods, it would be useful to state in the methods or results why TCR data was extracted from 6 donors when sequencing was performed on 9. It could be clarified in the figure legend for figure 3 that this represents the same sequencing data as in figure 2, just on a subset.
- 8) Figure 1 c – is the median or mean shown? Clarify in figure legend. Similarly, in this and other figures, is the viral load mean or median?
- 9) Figure 2 – in legend talks about parts e, f and g, however figure only has part e.
- 10) Figures 4 b and c: It would be helpful to show the median, as in figure 1b.
- 11) Figure 5c: clarify what the box and whiskers represent, ie 25th and 75th percentile and range?
- 12) Figures 6a vs 5d. In 5d MAIT cell function with *E. coli* stimulation goes up at viral setpoint, whereas in 6a it goes down. Which is correct?
- 13) Supplementary table 3. What does $n=?$
- 14) Supplementary table 4. The labelling of the different groups is not clear. For example, what does “Up V5 vs VA” refer to?
- 15) Supplementary methods:
 - a. “Chemokine receptor expression” rather than “chemokine expression”
 - b. These are generally quite repetitive of the material in the methods section and in some instances, eg RNASeq, do not necessarily add a great deal of additional detail. Could this be incorporated into the methods section?

Overall, this is a well conducted study. Studying dynamic changes in immune function in acute HIV infection is extremely challenging due to the difficulty identifying patients and obtaining serial samples, especially prior to infection. This study represents an important and valuable addition to the literature. It will influence future studies in the field.

Dr James Ussher

Department of Microbiology and Immunology

University of Otago

Response to reviewers for manuscript NCOMMS-19-06574 entitled "Dynamic MAIT cell response with progressively enhanced innateness during acute HIV-1 infection" by Lal et al. for publication in Nature Communications.

We would like to thank the reviewers for their positive comments and constructive criticisms that have helped us improve the manuscript. Point-by-point reply to the specific comments raised is listed below. The changes we made to the manuscript text in response to reviewers' comments are highlighted in **yellow** in the revised manuscript.

Reviewer 1

Overall comment: "Here the authors have, in intricate detail, studied the frequencies, functionality, phenotype, and transcriptional profiles of MAIT T cells in a very important cohort of HIV-infected individuals which were followed longitudinally after acquisition of HIV infection. The authors find that MAIT cells are highly activated during the acute phase of HIV infection, and then become reprogrammed towards innate-like functionality. Overall the experiments are well performed, the manuscript is well written, and the conclusions are supported by the data presented."

Response: We thank the reviewer for the overall very positive assessment of our manuscript.

Critique 1: "The authors should include discussion of the dynamics of MAIT cells in SIV-infected Asian macaques (Vinton JV, Juno, JI). The authors should also include a short discussion of how MAIT cells seem to not be influenced by TB infection of NHPs (Kaufman Inf. Imm.)."

Response: As suggested by the reviewer, we have now added a very brief paragraph in the Discussion addressing the findings by Juno et al and Vinton et al in two different species of non-human primates. We have opted to not discuss the TB study by Kauffman as we feel it does not pertain directly to our findings.

Critique 2: "Many of the observations have also been observed in other populations of CD8 T cells (PD1, HLADR, CD38 expression for example). If the authors gate on the CD8 T cells that are not MAIT cells, do expansions and activation of MAIT cells correlate with the other CD8 T cells? These data would point towards potential mechanisms underlying the observed phenomena."

Response: We agree with the reviewer that it is interesting to compare the MAIT cell response to acute HIV-1 infection, with that of the adaptive peptide-specific CD8 T cell compartment. The two T cell compartments are activated with similar kinetics as measured by induction of CD38 and HLA-DR. However the magnitude of the response in conventional CD8 T cells is much higher. For example, the fold change of HLA-DR at peak viral load in non-MAIT CD8 T cells compared to pre-infection is 8.4, while the fold change in MAIT cells at peak viral load compared to pre-infection is 2 (RFig. 1). Conventional CD8 T cells respond profoundly to systemic infection and HIV peptide antigens with strong adaptive clonal expansion in a peptide-specific manner. In contrast, MAIT cells respond to microbial riboflavin metabolites possibly as a consequence of impaired mucosal integrity and bacterial control during acute

infection. In fact, the difference in response magnitude between conventional CD8 T cells and MAIT cells supports the notion that the underlying mechanisms of activation are distinct.

RFig 1. Comparison of expression of activation markers in MAIT cells and conventional CD8 T cells in paired data from all patients in the study.

Critique 3: “The authors should mention the possible limitation of using V α 7.2 and CD161 in capturing a few non MAIT cells and how MR1 tetramers can also be used to identify MAIT cells.”

Response: In most experiments in the present manuscript we have used the V α 7.2+CD161+ definition of MAIT cells. In our experience this overlaps to 98-99% with the MR1 tetramer definition of MAIT cells. This we also show one example of in Figure 1a in the present manuscript. For most studies the V α 7.2+CD161+ definition of MAIT cells is sufficient and as good as the MR1 tetramer. This comes with one exception and that is when the minor CD4 subset of MAIT cells is considered, as some CD4+V α 7.2+CD161 T cells are non-MAIT cells. Therefore, in Figure 1e we use the MR1 tetramer to identify MAIT cells with the specific purpose to study the presence of subsets of MAIT cells. This methodological approach allows us to detect a relative decline in CD4+ MAIT cells during acute HIV infection.

Critique 4: “The discussion is quite verbose. This section could be shortened to focus on the biological significance of the observations and maybe novel therapeutic interventions which might be considered.”

Response: We have gone over the text and condensed some parts of the Discussion. With the addition of a brief discussion of non-human primate work, the Discussion is still the same length, just about four double-spaced pages. In general we feel the discussion is focused on the important findings and is suitable both in content and length.

Reviewer 2

Overall comment: “Lal et al. address important questions about the response of MAIT cells during acute HIV-1 infection. There have been numerous cross-sectional studies showing that MAIT cells are depleted in HIV+ individuals, compared to healthy controls. These studies have typically collected single time points from individuals with different phenotypes, but the HIV+ individuals have typically been infected for a long period of time. Therefore, there has not been an examination of how HIV impacts MAIT cell phenotypes longitudinally, beginning from

the time of acute HIV infection. Lal et al take advantage of access to samples from the RV217 study. In this study, there are paired longitudinal samples of HIV+ patients that were collected before and soon after becoming HIV-infected, so they can determine the impact that the early phase of HIV infection has on MAIT cell populations in humans. This manuscript provides an important first examination of this population, filling an essential gap in our understanding of MAIT cells in human HIV infection. As part of their study, these investigators do a really nice job of showing how the transcriptional profile of MAIT cells is highly dynamic during acute HIV infection, even though there is not depletion of MAIT cells during this time. This suggests that the ongoing pathogen response in the host is affecting MAIT cells, even if there are no apparent effect on the number and function of MAIT cells until chronic infection. It is likely that there is some intracellular reprogramming during acute infection that has long lasting impacts on the function of these cells in the host during chronic HIV. Overall, this paper is unique and helps the scientific community rethink the complex impact that HIV infection has on MAIT cells. Still, there are a few ways in which the data is presented that need clarification.”

Response: We thank the reviewer for the overall very positive assessment of our manuscript.

Critique 1: “In figure 1d, they show that HIV+ individuals have a higher frequency of MAIT cells in the gut mucosa. This could be skewed because the individuals will have a lower percentage of CD4 T cells, and thus a higher percentage of CD8 T cells. They should report this data in terms of the percent of MAIT cells as a fraction of CD8 T cells, rather than as a fraction of total T cells. By normalizing to the percent of CD8 T cells, they will have accounted for an overall increase in CD8+ T cell frequency post-HIV infection. In the recent paper by Juno et al (J Immunol), they also observe an increased MAIT cell frequency in the rectum, but it is displayed as a percent of CD8 T cells, rather than CD3 T cells. Given some of the similar observations made during acute SIV infection in the Juno et al paper, it is surprising that the authors do not examine a4b7 expression in their samples, nor do they include mention of that paper in their discussion.”

Response: We thank the reviewer for these suggestions. We have retrieved data on rectal biopsy size (in grams) and calculated the absolute number of MAIT cells per gram of tissue. This analysis shows a similar increase in acute infection as the data expressed as % of T cells, and we have added this new data as a new Figure 1f. Regarding a4b7, it is well known that MAIT cells in humans consistently express this receptor. We have not stained for a4b7 in this project, so comparisons with the increase in its expression in macaque MAIT cells are not possible.

Critique 2: “The TCR sequencing data is not clear. Methods explaining how this was performed are not readily apparent, except to say that it was derived from the RNA-seq data. The gene usage is also not described in the main text, even though other manuscripts (e.g. Gherardin et al) include which TRAV, TRAJ, and TRBV genes are being identified. They only show this sort of data briefly in supplemental figure 3. Furthermore, while a p-value is provided to say there is an increase in the number of unique alpha and beta clones, the statistical tests used to determine this increase is not explained. This lies in stark contrast to the detail presented in another recent paper (Howson et al) who show that TCR diversity of MAIT cells was increased after Salmonella infection. Lal et al include the TCR data to suggest that HIV-induced microbial translocation has led to the diversification of the TCR, but the information on how this assay was performed and interpreted is incomplete. There is no expectation to repeat this assay,

but it needs to be further explained and the authors need to ensure that any conclusions they make are not overstatements beyond what the data is showing.”

Response: We thank the reviewer for pointing this out. We have added more detail to the methods description of TCR sequence determination and relative abundance from the RNA-seq data in the supplementary methods section and in the supplementary materials file. We also included a table in the supplemental material (Supp Table 6) exploring the sequence and relative frequency of the most abundantly used TCR α and β chain clones in the 6 donors that were interrogated for TCR sequencing.

Critique 3: “The data in Figures 4e, 4f, and 4g are unclear. In 4e, they label the unstimulated and stimulated samples with different colors, but it is hard to identify these data in the plots. If distinguishing between the unstimulated and stimulated samples is important, then these plots for the different conditions should be shown separately. Further, is the data shown in figure 4E actually an example of the data presented in figure 4F? Lastly, Figure 4g shows no difference in the production of IFN γ between the CD56 $^-$ and CD56 $^+$ cells before and after HIV infection, which stands in stark contrast to the comment in the results that suggests a ‘progressive upregulation of innate characteristics in MAIT cells throughout acute HIV-1 infection’. Is the argument that there are more CD56 $^+$ MAIT cells after HIV infection, and that CD56 $^+$ cells are simply more innate-like? If so, that needs to be made clear.”

Response: We agree with the reviewer that the data presentation in Figure 4e, f and g in the original submission was unclear. In essence, the reviewer’s interpretation was correct. CD56 $^+$ MAIT cells respond more strongly to IL-12/18 stimulation. This enhanced responsiveness is retained in HIV infected subjects. Consequently, as the CD56 $^+$ subset of MAIT cells progressively expands there is an increased capacity in the MAIT cell compartment to respond to this innate cytokine stimulus. In the revised Figure 4 we have clarified the data presentation in panel 4e, moved the previous Figure 4f to the Supplement (Supplementary Fig. 4), and kept the previous Figure 4g as the new Figure 4f.

Critique 4: “The functional data presented in Figure 5 requires further explanation. In the text, they state that there is enhanced cytotoxic capacity of MAIT cells at the time of viral load set point, but the actual time point is not stated. Is this day 43? If so, how was significance determined? Was a statistical test performed on groups of samples, or did they compare the MAIT cell function of individual subjects over time? One of the major powers of this study is that they are following a longitudinal cohort. They should take advantage of this, where possible. If there is a reason they are not showing longitudinal data from the same individuals in Figure 5, then the reason for this should be described. In addition, figure 5d uses the value of ‘total function’ on the Y-axis, but it is not clear what ‘total function’ actually represents. This also needs to be clarified.”

Response: We agree with the reviewer that the data presentation in the previous Figure 5 was unclear in that it did not show the longitudinal nature of the data in a good way. In the revised version of Figure 5 we better display the longitudinal data and clearly state that changes compared to pre-infection time points are studied. We have also clarified both in the figure and in the legend that “total function”, means the percent of MAIT cells expressing at least one of the four functions studied. Statistical significance of functional changes was determined using the Friedman test with Dunn’s multiple comparison test. This information was added to the statistics section of the paper.

Reviewer 3

Overall comment: “Previous studies have shown that MAIT cells are depleted in HIV infection with reduced functionality. This has been evident as early in infection as 61 (39–86) weeks post infection (reference 25). The basis for this depletion, however, is unknown. Furthermore, the dynamics during acute infection remain to be defined. In this study, the authors have assessed changes in the MAIT cell population very early in HIV infection. Using samples from the RV217 cohort (in which high risk participants were screened twice weekly by NAT for infection), they were able to compare MAIT cell frequency, phenotype, and function at various timepoints post infection to pre-infection samples; post-infection they focused on peak viral load (median 16 days since first positive HIV-1 RNA), set point viral load (median 43 days), and early chronic infection (median 85 days). They also examined the frequency (but not absolute number or phenotype) of MAIT cells in sigmoid colon biopsies from untreated patients with acute infection from the RV254/SEARCH 010 study. They report that:

- 1) MAIT cells in blood expand early in infection, as measured by frequency and absolute number (albeit a small change), before declining in early chronic infection.
- 2) there is an increase in double negative MAIT cells as infection progresses
- 3) the frequency of MAIT cells amongst T cells in the sigmoid colon increases in acute infection
- 4) there is increased transcription and expression of genes associated with activation, exhaustion (including IRF4), and proliferation (Ki67) in acute infection
- 5) the expression of some of these markers of activation, exhaustion, and proliferation inversely correlate with MAIT cell frequency and count
- 6) the transcriptome of MAIT cells varies at different time points after infection, with evidence of increased “innateness” with increasing time from infection
- 7) the diversity of MAIT TCR sequences increases with time from infection, suggesting antigenic stimulation and proliferation of different clones
- 8) the CD56+ MAIT cell population increases as a proportion of MAIT cells and that this subpopulation retains functionality in early chronic infection
- 9) that after an early increase (at set point viral load) in degranulation (CD107a) and granzyme B expression following stimulation, at later timepoints there is a decline in function, including cytokine production
- 10) that markers of microbial translocation and inflammation correlate with MAIT cell activation and cytokine production.

These findings are novel and increase our understanding of the fate of MAIT cells in early HIV infection. They will be of interest to those in both the HIV and MAIT cell fields. Overall, they support a model of increased activation, proliferation, and trafficking of MAIT cells to the gut early in infection, driven by microbial translocation, followed by increasing exhaustion and loss of function with time from infection. It remains to be determined whether similar changes are seen in MAIT cells in tissue as chronic infection progresses – this should be examined in future studies. Future studies should also examine the impact of early initiation of ART on MAIT cell abundance, phenotype, function, and tissue distribution.”

Response: We thank the reviewer for the overall very positive assessment of our manuscript.

Critique 1: “Statistical analysis. Multiple Wilcoxon Signed Rank tests have been used to compare median values between selected pairs when there are >2 groups. This does not allow for multiple comparisons and increases the risk of a type 1 error. Kruskal-Wallis or Friedman tests would allow for the multiple comparisons and be more appropriate.”

Response: We agree with the reviewer that it is important to use the appropriate statistical test. However, in this study we do not have several groups, but rather repeated measures within the same group. We have redone all statistical analysis of longitudinal data in figures 1, 4 and 5 using the nonparametric Friedman test with the Dunn's multiple comparison test to adjust for repeated measures. Overall, the patterns remain the same despite that a few borderline significant p-values have now become non-significant after adjustment. We would however argue that in an exploratory study of a truly unique cohort and samples set one should not over-interpret p-values but rather focus in the broad patterns. Nevertheless, the main conclusions of the paper are still in the revised version of the paper very well supported by statistical significance.

Critique 2: "There are marked changes in T cells in the gut in early HIV infection (loss of CD4 T cells, increased numbers of CD8 T cells), which complicates the interpretation of the increased frequency of MAIT cells amongst T cells. It is unknown whether the absolute number of MAIT cells in the gut is changed."

Response: The authors agree with this important point and therefore we have retrieved data on rectal biopsy weight to calculate and present the number of MAIT cells per gram of tissue (new Figure 1f). This analysis shows a similar increase in acute infection as the data expressed as % of T cells. We can therefore state that there is an absolute increase of MAIT cells in the tissue, and we have added this new data as a new panel to Figure 1f.

Critique 3: "Method of quantitation of qPCR data. Gene expression levels are presented as Et, which = # of PCR cycles -Ct. This does not allow for differences in PCR efficiency or sample input (although 100 cells were sorted per well). Why was relative expression to housekeeping genes not performed?"

Response: For our multiplexed qPCR data from the Biomark platform we are using the protocols developed by Mario Roederer and colleagues (Dominguez et al, J. Immunol. Meth., 2013). Given that the individual primer sets are rigorously qualified, measurements of expression levels are highly reproducible when based on a fixed number of cells (in our case 100 cells). In fact, "normalization" based on housekeeping genes is not recommended when the data is based on a low number of cells, because the expression of such housekeeping genes varies quite significantly between single cells. The "normalization" process would therefore in this situation introduce variability into the data set and impair the ability to detect longitudinal changes. We have therefore used the clean data directly from the assay in our analyses.

Critique 4: "Insufficient information is provided on the bioinformatic analysis of RNASeq data."

Response: We agree with the point that with multiple analysis pipelines available for interpretation RNA-seq data, transparency concerning bioinformatics methods is critical. Additional detail regarding the analysis of RNA-seq data has been added to the Supplementary Methods section.

Critique 5: “Do inflammation and microbial translocation help preserve MAIT cell function into chronic infection? What is their association with MAIT cell abundance/frequency?”

Response: This is an interesting question, but difficult to answer. We have seen correlations between microbial translocation marker sCD14 and activation and function at the contemporaneous time point at peak viremia and set point viral load. We have also seen inflammation (CRP) at the early time points predicts function and the later time points. However, if such predictive associations extend later into chronic infection we currently do not have sufficient data to say.

Critique 6: “Significant correlations have been shown with MAIT cell frequency but not with abundance (eg figure 1h-j) – does this mean correlations with abundance were not significant? Could this be shown in supplementary data or as a table? Similarly, for the correlations shown in figure 6B and C, what other correlations were examined that were not significant?”

Response: In general, we have investigated possible correlative associations between measures of MAIT cells and patient characteristics and we report all statistically significant and biologically meaningful correlations in the paper. It is difficult to include a description of non-existing correlations in the paper. For the interest of the reviewer we can mention that we saw no correlation between MAIT cell frequency, absolute count, subset frequency (CD4/CD8/double negative MAIT cells), expression of markers of activation, and production of cytotoxic molecules or cytokines in response to stimuli, versus plasma IL-6 and IFABP. Likewise, the above aspects of the MAIT cell compartment showed no correlation with HIV disease progression values of set point VL, peak VL, and absolute CD4 T cell counts. We have displayed all significant association between the MAIT cell compartment and soluble factors in Figure 6.

Minor issues:

Minor critique 1: “Page 6 – “declining frequency of CD8+ MAIT cells (... at consecutive timepoints)”. Not entirely clear what is meant by “consecutive timepoints”. Measurements appear to have been made at >3 timepoints post-infection. It is unclear what exactly has been compared.”

Response: In the revised manuscript this paragraph has been rewritten for clarity and to reflect updated statistical analysis.

Minor critique 2: “For the 3 post-infection time points a median is provided for the number of days post first detection of HIV RNA. What is the range?”

Response: The peak viral load time point ranges from day 14 to day 22 post first detection of HIV, the set point viral load time point ranges from day 31 to day 50, and the early chronic time point ranges from day 60 to day 126. This information has been added to the text describing the study subjects on page 16 of the manuscript.

Minor critique 3: “Figure 2c – Venn diagram. Text states 59 genes downregulated at peak viraemia, however there appear to be 72 on the diagram.”

Response: We thank the reviewer for spotting this text error, which has been corrected in the revised version of the manuscript.

Minor critique 4: “Figure 5d: decline in functionality at early chronic timepoint with E. coli is marked as significant on the figure but the text states $p=0.14$.”

Response: With the updated repeated measures statistics this decline is no longer statistically significant, and the manuscript has been updated to reflect that.

Minor critique 5: “Page 11 – “cytokine storm” in acute HIV-1 infection. This terminology seems too strong. A cytokine storm in other settings is associated with hypotension and risk of organ dysfunction (eg CAR T cell therapy). Could an alternative term be used?”

Response: This sentence has been amended with less strong wording.

Minor critique 6: “Page 15. Please provide a reference for cART improving MAIT cell antibacterial responsiveness later in chronic infection.”

Response: A reference for this statement has now been added.

Minor critique 7: “While included in the supplementary methods, it would be useful to state in the methods or results why TCR data was extracted from 6 donors when sequencing was performed on 9. It could be clarified in the figure legend for figure 3 that this represents the same sequencing data as in figure 2, just on a subset.”

Response: We thank the reviewer for this suggestion. We have added additional sentences both in the Methods and in Figure 3 legend as suggested by the reviewer.

Minor critique 8: “Figure 1 c – is the median or mean shown? Clarify in figure legend. Similarly, in this and other figures, is the viral load mean or median?”

Response: Median is shown unless otherwise stated, and we have amended the legends to indicate this.

Minor critique 9: “Figure 2 – in legend talks about parts e, f and g, however figure only has part e.”

Response: Figure panel labels have now been added to Figure 2 f and g.

Minor critique 10: “Figures 4 b and c: It would be helpful to show the median, as in figure 1b.”

Response: We feel the trends in Figures 4 b and c are so clear that to add a line indicating the median would not really help the reader. We have thus opted not to show it.

Minor critique 11: “Figure 5c: clarify what the box and whiskers represent, ie 25th and 75th percentile and range?”

Response: This figure has been revised in response to Reviewer 2 point 4, and now has a different design without box plots.

Minor critique 12: “Figures 6a vs 5d. In 5d MAIT cell function with E. coli stimulation goes up at viral setpoint, whereas in 6a it goes down. Which is correct?”

Response: Both are correct. The axis scale in the two figures are different, as figure 6 describes evolution of soluble plasma protein data over the 50 days of infection.

Minor critique 13: Supplementary table 3. What does n=?

Response: We have added n values to the legend of Supp Table 3. They are also listed here for the reviewer:

Lines 1-4 and 9-11, n= 17

Lines 5-8, n=20

Lines 12-16, n=10

Minor critique 14: “Supplementary table 4. The labelling of the different groups is not clear. For example, what does “Up V5 vs VA” refer to?”

Response: The labelling of different groups in Supplementary Table 4 has been corrected.

Minor critique 15: Supplementary methods:

- a. “Chemokine receptor expression” rather than “chemokine expression”
- b. These are generally quite repetitive of the material in the methods section and in some instances, eg RNASeq, do not necessarily add a great deal of additional detail. Could this be incorporated into the methods section?

Response: We agree with this observation and at the request by the reviewers in general to add more detail to the RNA-seq methods section, we have added considerable new detailed information to the RNA-seq methods section and moved it to the main text of the manuscript. Redundancy from the methods section of supplemental material has been removed.

Concluding remark: Overall, this is a well conducted study. Studying dynamic changes in immune function in acute HIV infection is extremely challenging due to the difficulty identifying patients and obtaining serial samples, especially prior to infection. This study represents an important and valuable addition to the literature. It will influence future studies in the field.

Response: We thank the reviewer for this insightful and positive comment.

REVIEWERS' COMMENTS:

Reviewer #1 (Remarks to the Author):

The authors have addressed the concerns raised by the reviewers.

Reviewer #2 (Remarks to the Author):

This is a resubmission of 'Dynamic MAIT cell response with progressively enhanced innateness during acute HIV-1 infection' by Lal et al.

The main focus of the manuscript is to describe how acute HIV infection affects MAIT cell frequency and function. The authors have access to a unique cohort of HIV-infected individuals, and they use this study cohort to try to answer key questions about MAIT cells. Before this, other studies have used cross sectional cohorts of HIV+ and HIV-naïve individuals to assess the impact of HIV-infection on MAIT cells. This manuscript does an outstanding job of examining the longitudinal impact of HIV infection on MAIT cells.

The original submission was well done, but a handful of criticisms were made. The investigators addressed most of the comments, as outlined below. There are, however, just a few minor points that should still be addressed.

1. On lines 103-105, they explain the data from Figure 1d. They say there is a range of 8-67 MAIT cells/ul in early infection, and then 4-191 MAIT cells/ul at day 43. I don't understand how this matches the data that is shown in figure 1d. At day 43, it looks like the maximum number of MAIT cells/ul is about 100. Please clarify.

2. Data shown in Figure 1F that quantifies the number of MAIT cells per gram of tissue in the gut mucosa is a very nice addition.

3. Additional details were added about TCR sequencing and analysis, as they describe the use of the MiXCR software. They also add more details on the analysis of individual TCR chain frequencies.

4. The authors further expand their explanation of how CD56⁻ and CD56⁺ MAITs respond to IL-12/IL-18 stimulation to produce IFN γ . The figures 4e and 4f are fine, but their text refers to an increase in the production of IFN γ by CD56⁺ cells, compared to CD56⁻ naïve cells. The authors should identify which figure they are referring to when describing this data, as this is not currently clear.

5. Lastly, Figure 5 is explained much more clearly in this revised version.

Shelby O'Connor

Reviewer #3 (Remarks to the Author):

The authors have adequately addressed all the points I raised in my initial review.

Minor points of clarification arising from the altered manuscript:

1) Figure 1h. It is unclear which marker the statistical comparisons refer to. There are multiple different markers shown on the same graph. Could these be colour coded in the same way as the markers to assist with interpretation?

2) Cytokine production in response to bacteria. The text says that cytokine production by MAIT cells in response to bacterial stimulation was rather stable during the first three time points evaluated, and showed a trend towards decline at the final timepoint (Figure 5b, lines 257-260). However, there seems to be a decrease early (~18 days) of both IFN γ (which then recovers) and TNF (with a persistent decline).

3) Figure 5b legend. The figure legend states that the “median of these markers at pre-infection is shown as a solid red or green line respectively”. However, this description appears to better describe the dotted line. The coloured lines appear to reflect the median relative expression at each time point.

Dr James Ussher, University of Otago

Response to final comments by reviewers for manuscript NCOMMS-19-06574 entitled "Dynamic MAIT cell response with progressively enhanced innateness during acute HIV-1 infection" by Lal et al. for publication in Nature Communications.

We would like to thank the reviewers for their positive evaluation of the manuscript. Point-by-point reply to the specific comments raised is listed below.

Reviewer 1

Overall comment: "The authors have addressed the concerns raised by the reviewers."

Response: We thank the reviewer once again for the positive assessment of our manuscript.

Reviewer 2

Overall comment: "The main focus of the manuscript is to describe how acute HIV infection affects MAIT cell frequency and function. The authors have access to a unique cohort of HIV-infected individuals, and they use this study cohort to try to answer key questions about MAIT cells. Before this, other studies have used cross sectional cohorts of HIV+ and HIV-naïve individuals to assess the impact of HIV-infection on MAIT cells. This manuscript does an outstanding job of examining the longitudinal impact of HIV infection on MAIT cells. The original submission was well done, but a handful of criticisms were made. The investigators addressed most of the comments, as outlined below. There are, however, just a few minor points that should still be addressed."

Response: We thank the reviewer for the overall very positive assessment of our manuscript.

Critique 1: "On lines 103-105, they explain the data from Figure 1d. They say there is a range of 8-67 MAIT cells/ul in early infection, and then 4-191 MAIT cells/ul at day 43. I don't understand how this matches the data that is shown in figure 1d. At day 43, it looks like the maximum number of MAIT cells/ul is about 100. Please clarify"

Response: We thank the reviewer for pointing out this mistake. In the new revised version this has been corrected.

Critique 2: "Data shown in Figure 1F that quantifies the number of MAIT cells per gram of tissue in the gut mucosa is a very nice addition."

Response: Yes we agree with the reviewer and thank you for suggesting this addition.

Critique 3: "Additional details were added about TCR sequencing and analysis, as they describe the use of the MiXCR software. They also add more details on the analysis of individual TCR chain frequencies."

Response: Yes this we did.

Critique 4: “The authors further expand their explanation of how CD56- and CD56+ MAITs respond to IL-12/IL-18 stimulation to produce IFN γ . The figures 4e and 4f are fine, but their text refers to an increase in the production of IFN γ by CD56+ cells, compared to CD56 naïve cells. The authors should identify which figure they are referring to when describing this data, as this is not currently clear.”

Response: Our data indicate that CD56+ MAIT cells produce more IFN γ than CD56- MAIT cells. This pattern is similar between HIV infected and uninfected donors, while the CD56+ MAIT cell subset is increasing in numerical terms in HIV infected donors. We have further clarified these aspects in the new revised version of the manuscript.

Critique 5: “Lastly, Figure 5 is explained much more clearly in this revised version”

Response: Thank you.

Reviewer 3

Overall comment: “The authors have adequately addressed all the points I raised in my initial review.”

Response: We thank the reviewer for the overall very positive assessment of our manuscript.

Minor issues:

Minor critique 1: “Figure 1h. It is unclear which marker the statistical comparisons refer to. There are multiple different markers shown on the same graph. Could these be colour coded in the same way as the markers to assist with interpretation?”

Response: The level of significance indicated in the figure 2a (previously figure 1h) refers to all markers measured, longitudinally in comparison with the baseline pre-infection time point. We have now clarified this in the legend to figure 2.

Minor critique 2: “Cytokine production in response to bacteria. The text says that cytokine production by MAIT cells in response to bacterial stimulation was rather stable during the first three time points evaluated, and showed a trend towards decline at the final timepoint (Figure 5b, lines 257-260). However, there seems to be a decrease early (~18 days) of both IFN γ (which then recovers) and TNF (with a persistent decline)”

Response: Given the divergent patterns at day 16 after infection, the apparent decline in median total functionality does not reach significance.

Minor critique 3: “Figure 5b legend. The figure legend states that the “median of these markers at pre-infection is shown as a solid red or green line respectively”. However, this

description appears to better describe the dotted line. The coloured lines appear to reflect the median relative expression at each time point.”

Response: We thank the reviewer for pointing out this mistake. In the new revised version this has been corrected.